DOI: 10.1038/s41467-018-06556-9　　OPEN

# ATR/Chk1 signaling induces autophagy through sumoylated RhoB-mediated lysosomal translocation of TSC2 after DNA damage

Mingdong Liu[1], Taoling Zeng[1], Xin Zhang[1], Chunyan Liu[1], Zhihui Wu[1], Luming Yao[1], Changchuan Xie[1], Hui Xia[1], Qi Lin[1], Liping Xie[1], Dawang Zhou ⬚ [1], Xianming Deng[1], Hong-Lin Chan[2], Tong-Jin Zhao[1] & Hong-Rui Wang[1]

DNA damage can induce autophagy; however, the underlying mechanism remains largely unknown. Here we report that DNA damage leads to autophagy through ATR/Chk1/RhoB-mediated lysosomal recruitment of TSC complex and subsequent mTORC1 inhibition. DNA damage caused by ultraviolet light (UV) or alkylating agent methyl methanesulphonate (MMS) results in phosphorylation of small GTPase RhoB by Chk1. Phosphorylation of RhoB enhances its interaction with the TSC2, and promotes its sumoylation by PIAS1, which is required for RhoB/TSC complex to translocate to lysosomes. As a result, mTORC1 is inhibited, and autophagy is activated. Knockout of *RhoB* severely attenuates lysosomal translocation of TSC complex and the DNA damage-induced autophagy. Reintroducing wild-type but not sumoylation-resistant RhoB into *RhoB*⁻/⁻ cells restores the onset of autophagy. Hence, our study identifies a molecular mechanism for translocation of TSC complex to lysosomes in response to DNA damage, which depends on ATR/Chk1-mediated RhoB phosphorylation and sumoylation.

---

[1] State Key Laboratory of Cellular Stress Biology, Innovation Center for Cell Biology, School of Life Sciences, Xiamen University, Fujian 361102, China. [2] Institute of Bioinformatics and Structural Biology, Department of Medical Sciences, National Tsing Hua University, Hsinchu 30013, Taiwan. These authors contributed equally: Mingdong Liu, Taoling Zeng, Xin Zhang. Correspondence and requests for materials should be addressed to T.-J.Z. (email: zhaotj@xmu.edu.cn) or to H.-R.W. (email: wanghr@xmu.edu.cn)

Autophagy is a self-eating process to remove dysfunctional or unnecessary proteins and organelles through lysosomal degradation pathway, and therefore plays an essential role in maintaining cellular homeostasis[1,2]. It is a tightly regulated process involving formation of double-membraned autophagosome and subsequent conversion to autolysosome by fusing with lysosome, in which the proteins and organelles are degraded[3]. The core machinery of autophagy is orchestrated by a set of conserved proteins encoded by autophagy-related (ATG) genes, among which the Atg1/unc-51-like kinase (ULK) kinase complex functions as the most upstream component that is critical for initiating autophagosome formation[4,5]. The mammalian target of rapamycin (mTOR) complex 1 (mTORC1), a master regulator of cellular metabolism, inhibits initiation of autophagy by phosphorylating ULK1 and autophagy/beclin 1 regulator 1 (AMBRA1) to prevent activation of ULK1[6]. In addition, the tuberous sclerosis complex (TSC) complex (TSC complex) is the only known GTPase activating protein (GAP) for the small GTPase Rheb, a direct activator of mTORC1, and therefore functions as a key negative regulator for mTORC1 activity[7]. It has been shown that TSC complex is recruited to membrane of lysosomes to regulate Rheb activity, however, the molecular mechanism for lysosomal membrane recruitment of TSC complex needs to be further elucidated[8–10].

DNA damage can trigger various cellular responses including DNA repair, cell cycle arrest, senescence, apoptosis, necrosis, and autophagy. The outcome depends on types and severity of the damage, and death threshold determined by DNA damage response (DDR) pathways[11–13]. The ataxia–telangiectasia mutated (ATM) and ATM-related and RAD3-related (ATR) kinases compose ATM/Chk2 and ATR/Chk1 signaling cascades, respectively, which are the two major pathways in DDR. ATM/Chk2 pathway is activated by double-strand DNA breaks, whereas ATR/Chk1 signaling is usually activated by single-stranded DNA or bulky DNA lesions[14,15]. It has been reported that DDR signaling can activate autophagy through inhibiting mTORC1 activity via several pathways involving ATM, p53, AMP-activated protein kinase (AMPK), c-Jun N-terminal kinase (JNK), and poly (ADP-ribose) polymerase-1 (PARP1), etc.[12,16–23]. However, the underlying mechanism for DDR-induced autophagy, especially how ATR/Chk1 signaling triggers autophagy, remains elusive.

The Rho family small GTPases are key regulators of actin cytoskeleton[24,25]. Notably, RhoB exerts distinct roles in various biological contexts compared with its highly homologous family member RhoA and RhoC, although they are all involved in regulating actin dynamics and migration[26,27]. RhoB is a short-lived protein and its levels are frequently found decreased in many types of cancer[26,28]. Increased RhoB induces apoptosis in diverse cancer cell lines[29–34], while knockout of *RhoB* significantly inhibits DNA damage-induced apoptosis[35], suggesting that RhoB suppresses tumorigenesis through promoting cell death. Our previous study showed that E3 ubiquitin ligase Smurf1 targets RhoB for degradation to maintain a relative low RhoB level in the basal state. Activation of ATR/Chk1 signaling upon DNA damage induces self-degradation of Smurf1, and therefore prevents RhoB from Smurf1-mediated degradation[36].

In this study, we found that RhoB is phosphorylated by Chk1 after DNA damage, which promotes its binding to SUMO E3 ligase PIAS1 and subsequent sumoylation. Meanwhile, this phosphorylation also enhances the binding of RhoB to TSC complex. Therefore, the sumoylated phospho-RhoB functions as a carrier protein to translocate TSC complex to lysosomes, initiating autophagy by inhibiting mTORC1 activity.

## Results

**PIAS1 mediates sumoylation of small GTPase RhoB.** Our previous study showed that Smurf1 targets RhoB for ubiquitination to control its abundance in cells under basal conditions. Upon DNA damage, ATR/Chk1 signaling triggers Smurf1 self-degradation and leads to an accumulation of RhoB to promote apoptosis[36]. To further investigate the role of RhoB in DDR, we carried out a yeast-two-hybrid screen using RhoB as the bait to identify novel RhoB-binding proteins. Interestingly, we found that among the identified candidates several clones encode ubiquitin-conjugating enzyme 9 (Ubc9), the only known SUMO-conjugating E2 enzyme in mammalian cells. To verify this interaction, we performed coimmunoprecipitation assay (Fig. 1a) and in vitro GST pull-down assay (Supplementary Fig. 1a), confirming that RhoB interacts with Ubc9 in cells and in vitro.

We therefore examined whether RhoB could be sumoylated. We immobilized His-tagged RhoB using Nickel–nitrilotriacetic acid (Ni-NTA) agarose beads followed by immunoblotting to detect the conjugation of SUMO. Indeed, we found that RhoB could be sumoylated with a preference for SUMO2 conjugation, and coexpression of Ubc9 enhanced RhoB sumoylation (Fig. 1b). In addition, we carried out in vitro sumoylation assay and confirmed that Ubc9 could directly target RhoB for SUMO2 conjugation (Supplementary Fig. 1b).

We next examined the effects of PIAS family of SUMO E3 ligases on RhoB sumoylation. As shown in (Fig. 1c), PIAS1 significantly enhanced the sumoylation of RhoB, whereas other PIAS family members did not. Meanwhile, knockdown of endogenous PIAS1 remarkably inhibited sumoylation of RhoB (Fig. 1d), indicating that PIAS1 is a major SUMO E3 ligase for RhoB. In addition, overexpression of wild-type PIAS1 but not PIAS1-C351S, a catalytic inactive mutant, promoted RhoB sumoylation (Fig. 1e). Similarly, wild-type PIAS1 but not PIAS1-C351S increased SUMO-conjugation in the in vitro sumoylation assay (Supplementary Fig. 1c), indicating that PIAS1-mediated increase of RhoB sumoylation is dependent on the catalytic activity of PIAS1. We further confirmed that PIAS1 could interact with RhoB in cells by trapping endogenous and exogenous RhoB using catalytically inactive PIAS1-C351S (Fig. 1f; Supplementary Fig. 1d). Moreover, in vitro GST pull-down assay indicated a direct interaction between PIAS1 and RhoB (Supplementary Fig. 1e).

To identify the sumoylation site(s) on RhoB, we carried out in vitro sumoylation reaction of RhoB to obtain sumoylated RhoB for mass spectrometry analysis. SUMO-conjugation could be detected on three lysine residues (Lys7, Lys135, and Lys194) in RhoB (Supplementary Fig. 1f). Mutations of Lys6 and 7, Lys135, or Lys194 to arginine (K6, 7R, K135R, or K194R) significantly decreased sumoylation levels of RhoB, and mutation of all the four lysine residues to arginines (4KR) nearly completely abolished the RhoB sumoylation in cells and in vitro (Fig. 1g; Supplementary Fig. 1g). Accordingly, PIAS1 could only enhance sumoylation of wild-type RhoB but not RhoB-4KR (Fig. 1h). Of note, RhoB-4KR and wild-type RhoB showed similar binding affinity to the Rho-binding domain (RBD) of Rho effector protein Rhotekin (Supplementary Fig. 1h), and similar effect on enhancing cell stress fiber formation (Supplementary Fig. 1i), indicating that RhoB-4KR is folded properly and able to activate ROCK pathway.

**Sumoylation is required for RhoB translocation to lysosomes.** We next explored the upstream signal(s) that can trigger sumoylation of RhoB in cells. Interestingly, although treatment of ultraviolet (UV), alkylating agent methyl methanesulphonate (MMS), topoisomerase I inhibitor camptothecin (CPT), or

topoisomerase II inhibitor doxorubicin (DOX) all caused severe DNA damage (Supplementary Fig. 2a), only UV and MMS significantly enhanced RhoB sumoylation, indicating a specific effect of UV or MMS treatment on regulating RhoB sumoylation (Fig. 2a). Consistent with our previous report[36], UV or MMS treatment strongly activated Chk1, whereas CPT and DOX treatment mainly activated Chk2 (Supplementary Fig. 2b).

To examine whether endogenous RhoB in cells could be sumoylated, we exogenously expressed His-tagged SUMO2 in cells, pull-down His-SUMO2 using Ni-NTA agarose beads, and immunoblotted RhoB to detect the conjugation of RhoB and SUMO2. Strikingly, sumoylation of endogenous RhoB could only be detected after cells were treated with UV or MMS, and, as

indicated, the major conjugation form is mono-sumoylated RhoB (Fig. 2b). Of note, the levels of endogenous RhoB dramatically increased in cells treated with UV or MMS (Fig. 2b), which is in good agreement with our previous study[36]. It is noteworthy that UV or MMS treatment significantly enhanced sumoylation of exogenous His-tagged RhoB, even though the protein levels of His-RhoB were adjusted to a similar level (Supplementary Fig. 2c), indicating that the upregulation of endogenous RhoB sumoylation is not only due to the increase of RhoB protein levels, but also because of an increased efficiency of sumoylation. As expected, the 4KR mutation totally abolished the UV or MMS-induced RhoB sumoylation (Supplementary Fig. 2c).

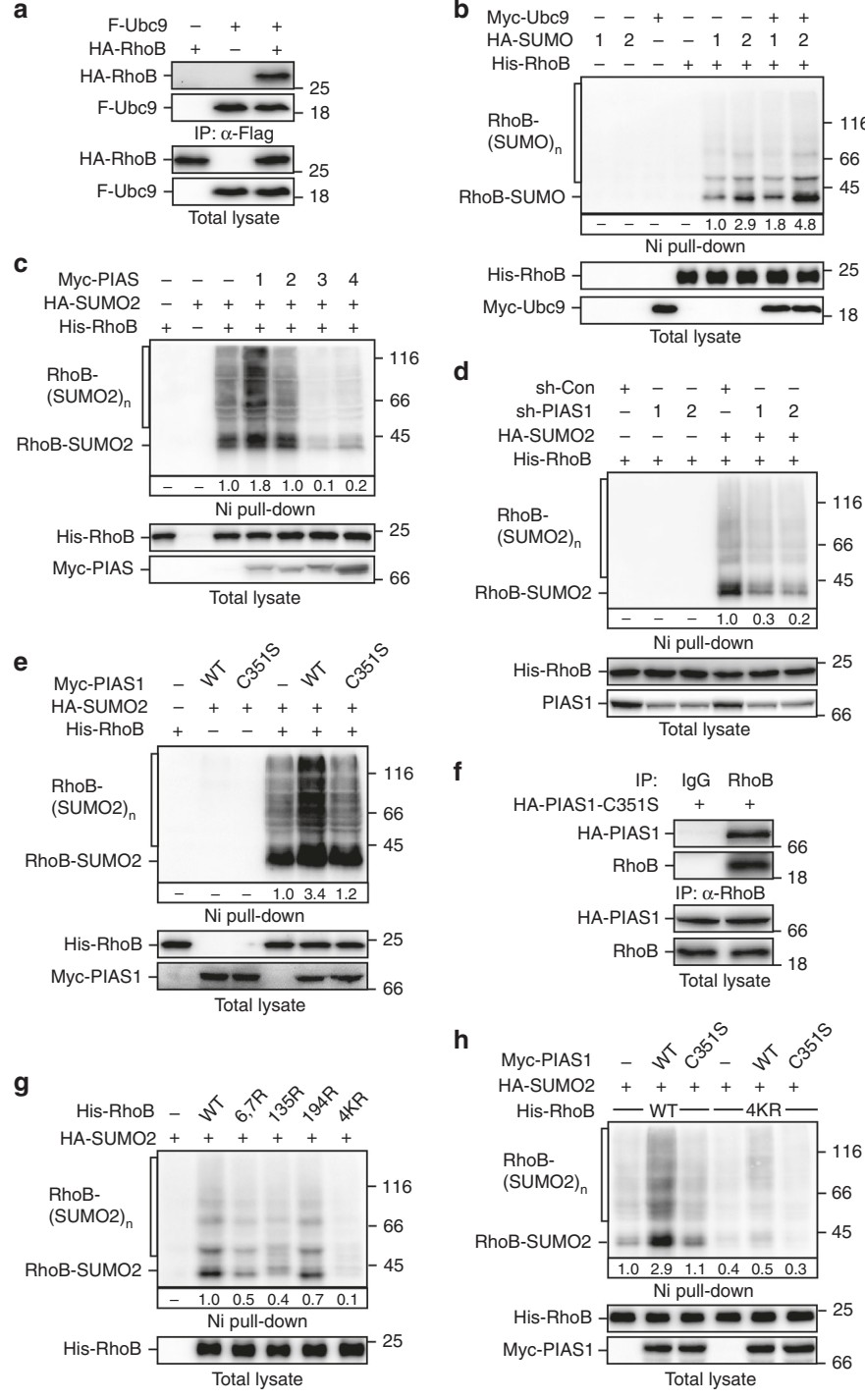

**Fig. 1** RhoB is sumoylated by PIAS1. **a** RhoB interacts with Ubc9. HEK293T cells transfected with indicated combinations of Flag-tagged Ubc9 (F-Ubc9) and HA-tagged RhoB (HA-RhoB) were subjected to anti-Flag immunoprecipitation (IP) followed by immunoblotting assay to detect associated RhoB. **b** RhoB is sumoylated. HEK293T transfected with indicated combinations of His-tagged RhoB (His-RhoB), HA-tagged SUMO (HA-SUMO) 1 or 2, and Myc-tagged Ubc9 (Myc-Ubc9) were lysed with 6 M guanidine-HCl followed by Ni-NTA agarose beads pull-down (Ni pull-down) assay. SUMO-conjugated RhoB was detected by immunoblotting with anti-HA. Conjugation of mono-SUMO and multi-SUMO to RhoB are indicated as RhoB-SUMO and RhoB-(SUMO)$_n$, respectively. **c** PIAS1 promotes sumoylation of RhoB. HEK293T cells cotransfected with His-RhoB, HA-SUMO2, and indicated Myc-tagged PIAS (Myc-PIAS) family member 1–4 were subjected to sumoylation assay as described in panel **b**. **d** Knockdown of PIAS1 attenuates RhoB sumoylation. HEK293T cells transfected with indicated combinations of His-RhoB, HA-SUMO2, and shRNAs against PIAS1 were subjected to sumoylation assay as described in panel **b**. **e** The E3 catalytic activity is required for PIAS1-mediated RhoB sumoylation. HEK293T cells were transfected with His-RhoB, HA-SUMO2, and Myc-PIAS1 wild-type (WT) or catalytically inactive mutant (C351S) as indicated. The cells were subjected to sumoylation assay as described in panel **b**. **f** PIAS1 interacts with endogenous RhoB. HeLa cells transduced with lentivirus encoding HA-tagged PIAS1-C351S mutant (HA-PIAS1-C351S) were subjected to anti-RhoB IP followed by immunoblotting with rat anti-HA to detect associated HA-PIAS1-C351S. **g** Sumoylation sites mapping on RhoB. HEK293T cells transfected with HA-SUMO2 and indicated His-RhoB mutants were subjected to sumoylation assay as described in panel **b**. **h** PIAS1 enhances sumoylation of WT but not 4KR RhoB. HEK293T cells transfected with indicated combinations of HA-SUMO2, Myc-PIAS1 (WT or C351S), and His-RhoB (WT or 4KR) were subjected to sumoylation assay as described in panel **b**

We found that UV or MMS treatment dramatically changed the subcellular localization of RhoB. In cells without treatment, RhoB mainly localized to plasma membrane and some small vesicles. Specifically, in cells treated with UV or MMS, endogenous RhoB exhibited a strong aggregation pattern and colocalized with LAMP1 (lysosome-associated membrane protein 1) (Fig. 2c; Supplementary Fig. 2d), indicating that RhoB could be translocated to lysosomes after UV or MMS treatment. In good agreement with that RhoB sumoylation occurs specifically in response to UV or MMS treatment (Fig. 2a), treatment with CPT or DOX did not alter the plasma membrane localization of RhoB (Fig. 2c; Supplementary Fig. 2d). It is noticeable that although dissociated from plasma membrane, RhoB-4KR was not translocated to lysosomes (Fig. 2d; Supplementary Fig. 2e). Similarly, knockdown of PIAS1 did not block the dissociation of RhoB from plasma membrane after UV or MMS treatment, but significantly impeded the translocation of RhoB to lysosomes (Supplementary Fig. 2f, g), indicating that the lysosomal translocation of RhoB is dependent on the sumoylation.

**Sumoylation of RhoB is required for UV/MMS-induced autophagy**. Because lysosomes play essential roles in the process of autophagy, we hypothesized that RhoB might be involved in regulating autophagy after DNA damage. To evaluate this, we first detected the formation of LC3 puncta by fluorescence microscopy using mCherry red fluorescent protein-tagged LC3 (mRFP-LC3). Indeed, knockout of *RhoB* markedly decreased number of mRFP-LC3 puncta in cells treated with UV or MMS but not in cells treated with CPT or DOX (Fig. 3a; Supplementary Fig. 3a). In line with the microscopy assay, knockout of *RhoB* significantly attenuated the upregulation of LC3-II (autophagosome-associated form of LC3) and downregulation of p62/SQSTM1 in cells treated with UV or MMS but not CPT or DOX, and the presence of lysosome inhibitor chloroquine increased the LC3-II levels under these conditions (Fig. 3b; Supplementary Fig. 3b–d), indicating that RhoB positively affects autophagic flux in response to UV or MMS treatment. Reintroducing wild-type RhoB but not sumoylation-resistant mutant RhoB-4KR into *RhoB*$^{-/-}$ cells restored the UV or MMS-induced LC3 aggregation (Fig. 3c; Supplementary Fig. 3e). Accordingly, reintroduction of wild-type RhoB but not RhoB-4KR upregulated LC3-II and downregulated p62/SQSTM1 (Fig. 3d; Supplementary Fig. 3f). Moreover, knockdown of PIAS1 significantly blocked UV or MMS-induced autophagy (Supplementary Fig. 3g–j), confirming that sumoylation is required for UV or MMS-mediated autophagy.

We next further evaluated the function of RhoB in regulating autophagic flux using tandem fluorescent protein-tagged LC3

(mRFP-GFP-LC3) to monitor maturation of autophagosomes into autolysosomes. In this assay, the fluorescence of GFP is quenched by the low pH inside autolysosomes, whereas mRFP is more resistant to the acidic condition[37]. Hence, both GFP and RFP fluorescence will exhibit in autophagosomes (yellow puncta), but only mRFP fluorescent signal can be detected in autolysosomes (red puncta). Indeed, knockout of *RhoB* remarkably decreased the numbers of both red and yellow puncta in cells after UV or MMS treatment (Fig. 3e; Supplementary Fig. 3k), indicating that the formation of both autophagosomes and autolysosomes is inhibited. Consistently, reintroduction of wild-type RhoB but not RhoB-4KR in *RhoB*$^{-/-}$ cells restored both red and yellow puncta formation upon UV or MMS treatment (Fig. 3f; Supplementary Fig. 3l), suggesting that sumoylated RhoB is required for promoting autophagic flux after DNA damage caused by UV or MMS.

To further verify the role of RhoB in regulating DNA damage-induced autophagy, we performed electron microscopy to examine the formation of autophagosomes and autolysosomes in *RhoB*$^{+/+}$ and *RhoB*$^{-/-}$ cells. In good agreement with the fluorescence microscopy assay, knockout of *RhoB* dramatically decreased the numbers of both autophagosomes (double-membraned vesicles) and autolysosomes (single-membraned vesicles) in cells treated with UV or MMS (Fig. 3g; Supplementary Fig. 3m). Furthermore, reintroduction of wild-type RhoB but not RhoB-4KR rescued the formation of autophagosomes and autolysosomes upon UV or MMS treatment (Supplementary Fig. 3n, o). Hence, these results clearly demonstrated that sumoylated RhoB plays a pivotal role in mediating autophagy in response to UV or MMS treatment.

**Sumoylation of RhoB is critical for UV/MMS-induced mitophagy**. Because we observed a significant number of mitochondria enclosed in the autophagosomes and autolysosomes in cells treated with UV or MMS, we next explored the role of RhoB in regulating mitophagy in this scenario. For this end, we first compared mitochondria removal in control and *RhoB* knockout cells after UV or MMS treatment. Indeed, knockout of *RhoB* markedly blocked the removal of mitochondria in cells treated with UV or MMS, as shown by immunofluorescence (Fig. 4a) and immunoblotting analysis (Fig. 4b) for the mitochondrial heat-shock protein Hsp60. In addition, treatment with lysosome inhibitor chloroquine prevented the decrease of Hsp60 levels induced by UV or MMS (Fig. 4c), indicating that the loss of Hsp60 is through mitophagy. Consistently, reintroduction of wild-type RhoB but not RhoB-4KR reinstated the clearance of mitochondria after treatment with UV or MMS (Fig. 4d; Supplementary Fig. 4a).

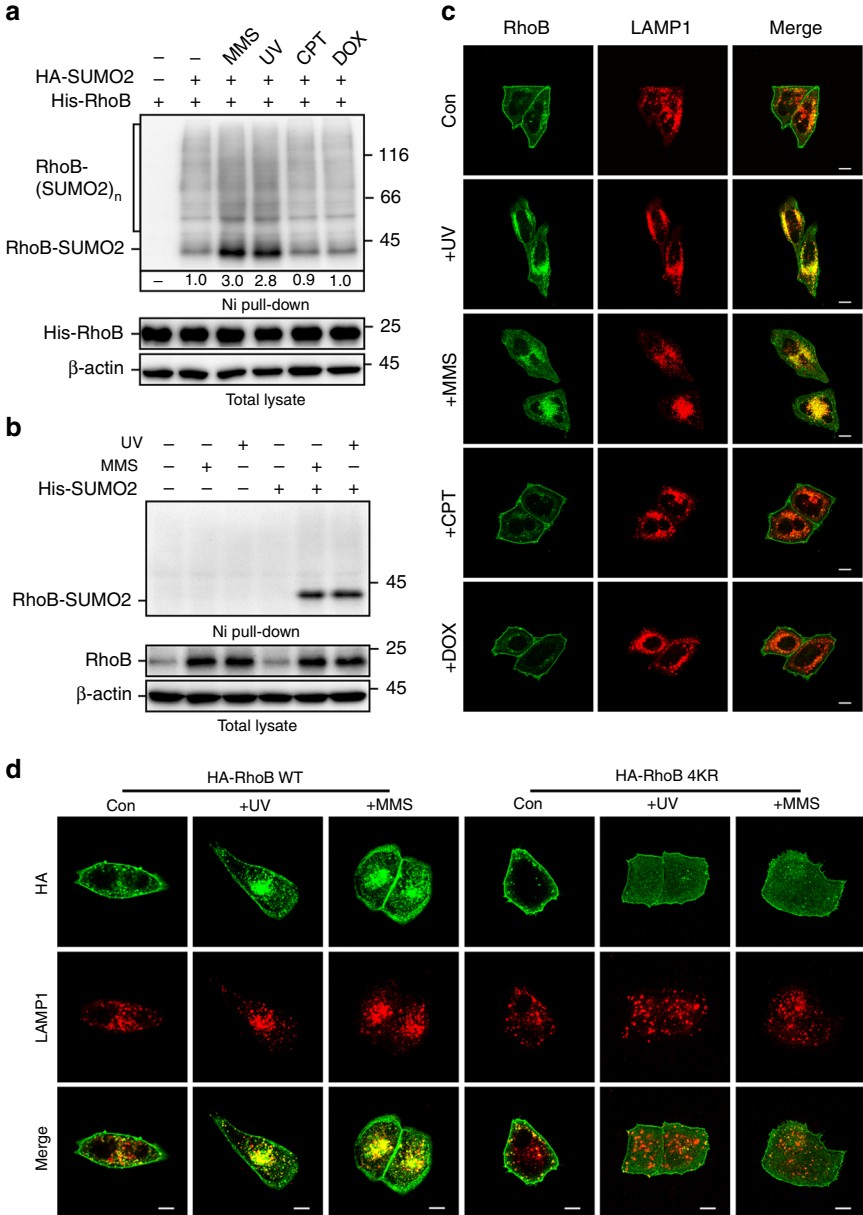

**Fig. 2** Sumoylated RhoB is translocated to lysosomes. **a** Sumoylation of RhoB is specific to ultraviolet (UV) or methyl methanesulphonate (MMS) treatment. One hour after treated with UV (80 Jm$^{-2}$) or 2 h after treated with MMS (0.5 mM), camptothecin (CPT) (10 μM), or doxorubicin (DOX) (0.5 μM), HEK293T cells expressing HA-SUMO2 and His-RhoB were subjected to sumoylation assay to detect the SUMO2 conjugation of RhoB. **b** UV or MMS treatment enhances sumoylation of endogenous RhoB. HeLa cells with expression of His-SUMO2 were subjected to sumoylation assay 2 h after treated with UV (80 Jm$^{-2}$) or 4 h after treated with MMS (0.5 mM). SUMO2 conjugation of RhoB was detected by immunoblotting with RhoB antibody. **c** Endogenous RhoB colocalizes with LAMP1 after UV or MMS but not CPT or DOX treatment. HeLa cells were subjected to immunofluorescence assay 2 h after treated with or without UV (80 Jm$^{-2}$), MMS (0.5 mM), CPT (10 μM), or DOX (0.5 μM). Scale bar, 10 μm. **d** RhoB WT but not RhoB-4KR colocalizes with LAMP1 after UV or MMS treatment. U2OS cells expressing HA-RhoB (WT or 4KR) were subjected to immunofluorescence assay 4 h after UV (80 Jm$^{-2}$) or MMS (0.5 mM) treatment to examine the localization of HA-RhoB and endogenous LAMP1. Scale bar, 10 μm

To further verify the effect of RhoB on turnover of mitochondria in response to UV or MMS-induced DNA damage, we performed double fluorescence assay using a previous reported mRFP-GFP-tagged signal peptide (residues 101–152) of mitochondrial fission 1 protein (mRFP-GFP-FIS1$_{101-152}$), which localizes to the outer membrane of mitochondria, to monitor the delivery of mitochondria to autolysosomes during mitophagy[38]. Similar to mRFP-GFP-LC3, mRFP-GFP-FIS1$_{101-152}$ displayed both red and green fluorescence in autophagosomes but only red signal in autolysosomes. In line with the results above,

knockout of *RhoB* remarkably blocked the red-only puncta formation in cells treated with UV or MMS (Figs. 4e, f), and reintroduction of wild-type RhoB but not RhoB-4KR restored the formation of red-only puncta (Supplementary Fig. 4b). Hence, our results demonstrated that sumoylation of RhoB is also necessary for UV or MMS-induced mitophagy.

Our previous study showed that RhoB is required for UV or MMS-induced apoptosis[36]; therefore, we further investigated whether sumoylated RhoB-mediated autophagy is required for the cell death. Consistent with our previous report, knockout of

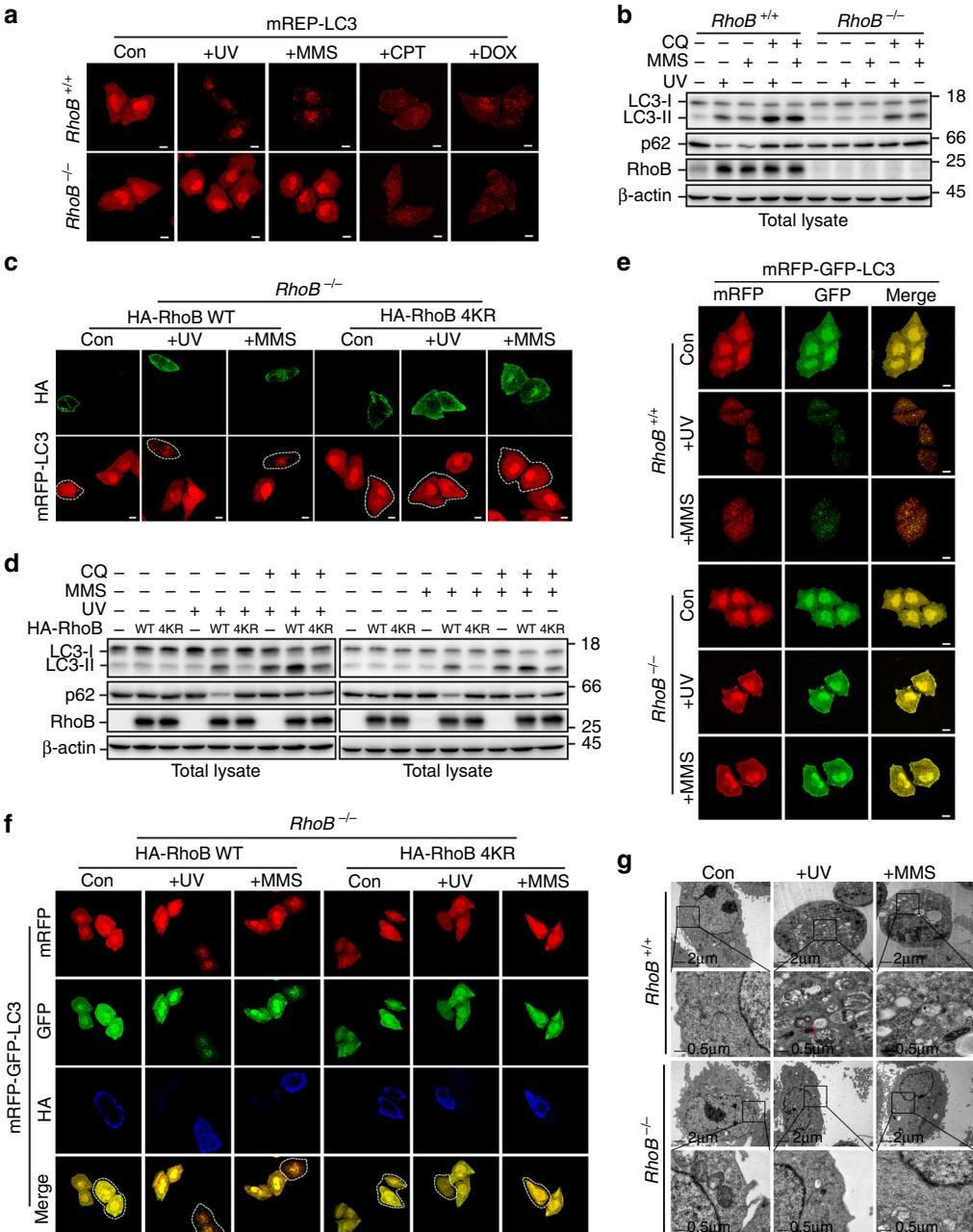

**Fig. 3** Sumoylated RhoB is required for UV or MMS-induced autophagy. **a** Knockout of *RhoB* decreases UV or MMS treatment-induced mRFP-LC3 puncta formation. *RhoB*[+/+] or *RhoB*[−/−] cells stably expressing mCherry red fluorescent protein fused LC3 (mRFP-LC3) were subjected to fluorescence microscopy 4 h after UV (80 Jm[−2]), 6 h MMS (0.5 mM), 8 h CPT (10 μM), or 8 h DOX (0.5 μM) treatment. Scale bar, 10 μm. **b** Knockout of *RhoB* inhibits UV or MMS-induced augment of LC3-II. *RhoB*[+/+] or *RhoB*[−/−] cells pretreated 1 h with or without 50 μM chloroquine (CQ) were subjected to immunoblotting 4 h after UV (80 Jm[−2]) or 6 h with MMS (0.5 mM) treatment. **c** Reintroduction of RhoB WT but not 4KR restores UV or MMS-induced mRFP-LC3 aggregation. *RhoB*[−/−] cells stably expressing mRFP-LC3 were transduced with lentivirus encoding HA-RhoB WT or 4KR. Cells were subjected to immunofluorescence assay 4 h after UV (80 Jm[−2]) or 6 h after MMS (0.5 mM) treatment. The dot lines indicate cells expressing RhoB WT or 4KR. Scale bar, 10 μm.
**d** Reintroduction of RhoB WT but not 4KR restores UV or MMS-induced augment of LC3-II. *RhoB*[−/−] cells transduced with HA-RhoB WT or 4KR were treated as in panel **b**. **e** Knockout of *RhoB* decreases the autophagic flux promoted by UV or MMS. *RhoB*[+/+] or *RhoB*[−/−] cells stably expressed mRFP-GFP-LC3 were subjected to fluorescence microscopy after 8 h UV (80 Jm[−2]) or 10 h MMS (0.5 mM) treatment. Scale bar, 10 μm. **f** Reintroduction of RhoB WT but not 4KR promotes the autophagic flux upon UV or MMS treatment. *RhoB*[−/−] cells stably expressing mRFP-GFP-LC3 were transduced with HA-RhoB WT or 4KR. Cells were treated and subjected to immunofluorescence assay as in panel **e**. The dot lines indicate cells expressing RhoB WT or 4KR. Scale bar, 10 μm. **g** Knockout of *RhoB* attenuates UV or MMS-induced autophagosomes and autolysosomes formation. *RhoB*[+/+] or *RhoB*[−/−] cells were assessed by electron microscopy 6 h after UV (80 Jm[−2]) or 8 h after MMS (0.5 mM) treatment. The magnified images are the areas indicated by the squares. Red arrow indicates autophagosome/autolysosome with mitochondria

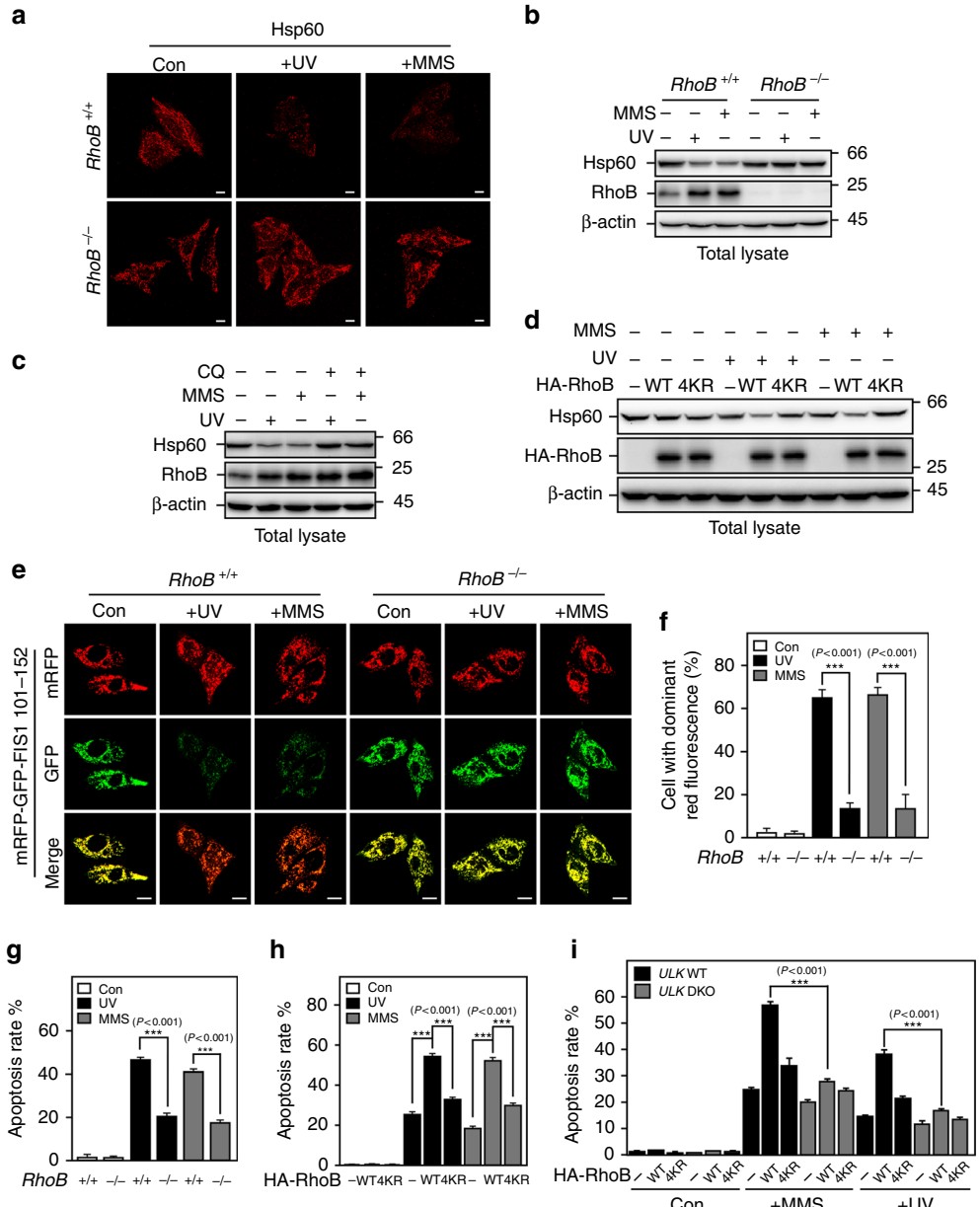

**Fig. 4** Sumoylated RhoB is required for UV or MMS-induced mitophagy and cell death. **a** and **b** Knockout of *RhoB* inhibits mitochondria clearance after UV or MMS treatment. *RhoB*$^{+/+}$ or *RhoB*$^{-/-}$ cells were treated as Fig. 3e and subjected to immunofluorescence (**a**) or immunoblotting (**b**). Scale bar, 10 μm. **c** UV or MMS-induced downregulation of Hsp60 is through lysosome pathway. HeLa cells pretreated 1 h with or without 100 μM CQ were treated and processed as in panel **b**. **d** Reintroduction of RhoB WT but not 4KR down-regulates Hsp60 after UV or MMS treatment. *RhoB*$^{-/-}$ cells transduced with HA-RhoB (WT or 4KR) were treated as in panel **b**. **e** and **f** Knockout of *RhoB* decreases UV or MMS-promoted mitophagic flux. *RhoB*$^{+/+}$ and *RhoB*$^{-/-}$ cells stably expressing mRFP-GFP-tagged signal peptide (residues 101–152) of mitochondrial fission 1 protein (mRFP-GFP-FIS1$_{101-152}$) were treated and processed as in panel **a**. Scale bar, 10 μm. The percentage of cells with dominant red fluorescence were quantified and plotted in panel **f**. Five random areas were counted for each experiment and data of three independent experiments were assessed by one-way ANOVA ($F_{(5,84)} = 174.51$) followed by LSD post hoc test after arcsine transformation and represented as mean ± SD (**f**). **g** and **h** Sumoylated RhoB is required for UV or MMS-induced apoptosis. *RhoB*$^{+/+}$ or *RhoB*$^{-/-}$ cells (**g**), or *RhoB*$^{-/-}$ cells transduced with RhoB WT or 4KR (**h**) were subjected to flow cytometry assay to determine the apoptosis rate 30 h after treated with UV (80 Jm$^{-2}$) or MMS (0.5 mM). Data of three independent experiments were assessed by one-way ANOVA ($F_{(5,12)} = 2939.3$) (**g**) or ($F_{(8,18)} = 4854.2$) (**h**) followed by LSD post hoc test after arcsine transformation and represented as mean ± SD. **i** Double knockout of *ULK1/2* abrogates RhoB-promoted apoptosis in response to UV or MMS treatment. *ULK1/2* WT or double knockout (DKO) MEF cells were transduced with RhoB. Cells were subjected to flow cytometry assay to determine the apoptosis rate 24 h after treated with UV (80 Jm$^{-2}$) or MMS (0.5 mM). Data of three independent experiments were assessed by one-way ANOVA ($F_{(17,36)} = 1069.3$) followed by LSD post hoc test after arcsine transformation and represented as mean ± SD

*RhoB* dramatically inhibited UV or MMS-induced apoptosis (Fig. 4g). Interestingly, reintroduction of wild-type but not 4KR RhoB rescued the apoptosis induced by UV or MMS treatment (Fig. 4h), suggesting that sumoylation is necessary for RhoB-mediated cell death. Moreover, double knockout of *ULK1/2* significantly attenuated RhoB-promoted apoptosis in response to UV or MMS treatment (Fig. 4i), indicating autophagy is required for RhoB to promote cell death upon DNA damage.

**ATR/Chk1 signaling is required for UV/MMS-induced autophagy**. As we have shown that UV or MMS-induced DDR is mainly through Chk1 signaling pathway (Supplementary Fig. 2b), we therefore investigated whether ATR/Chk1 signaling is also involved in regulating the UV or MMS-triggered autophagy. Indeed, treated with ATR inhibitor but not ATM inhibitor dramatically attenuated UV or MMS-induced puncta formation of mRFP-LC3 (Fig. 5a, b), suggesting that ATR but not ATM is a major regulator of UV or MMS-triggered autophagy. Consistently, inhibiting ATR but not ATM strongly blocked UV or MMS-mediated Chk1 activation (Supplementary Fig. 5a). The role of ATR in controlling UV or MMS-induced autophagy is also confirmed by knocking down ATR (Supplementary Fig. 5b–d).

We next examined the necessity of Chk1 in the UV or MMS-induced autophagy. As predicted, Chk1 but not Chk2 inhibitor drastically decreased the LC3 aggregation after UV or MMS treatment (Fig. 5c, d). In addition, knockdown of Chk1 also markedly attenuated LC3 puncta formation in cells treated with UV or MMS (Supplementary Fig. 5e–g). Moreover, Chk1 inhibitor significantly blocked the upregulation of LC3-II induced by UV or MMS treatment (Fig. 5e; Supplementary Fig. 5h). We further determined the role of Chk1 in regulating autophagic flux using mRFP-GFP-LC3 and found that knockdown of Chk1 activity dramatically diminished the formation of both red and yellow puncta in cells treated with UV or MMS (Fig. 5f; Supplementary Fig. 5i). Thus, these results indicated that ATR/Chk1 signaling pathway plays a key role in controlling UV or MMS-induced autophagy.

**Phosphorylation of RhoB is essential for its translocation**. To investigate whether ATR/Chk1 signaling-mediated autophagy is through a RhoB-dependent pathway, we first examined whether Chk1 activity is necessary for the translocation of RhoB to lysosomes. Treatment with Chk1 inhibitor or knockdown of Chk1 remarkably diminished the colocalization between RhoB and LAMP1 (Fig. 6a; Supplementary Fig. 6a–c), indicating Chk1 activity is required for lysosomal translocation of RhoB. Meanwhile, Chk1 inhibitor efficiently blocked both endogenous and exogenous RhoB sumoylation induced by UV or MMS (Fig. 6b; Supplementary Fig. 6d). The upregulation of RhoB by UV or MMS was also blunted when Chk1 was inhibited (Fig. 6b), which is consistent with our previous study[36]. Moreover, using kinase-dead mutant Chk1-D130A[39], we were able to detect its interaction with both exogenous and endogenous RhoB (Fig. 6c; Supplementary Fig. 6e), suggesting that Chk1 may directly target RhoB for phosphorylation.

Next, we performed in vitro phosphorylation assay and observed that Chk1 could phosphorylate RhoB on its threonine but not serine residue(s) by using phospho-Thr and phospho-Ser specific antibodies (Fig. 6d). We identified that Thr173 and Thr175 of RhoB was phosphorylated by Chk1 using matrix-assisted laser desorption/ionization time-of-flight mass spectrometry (MALDI-TOF-MS) (Supplementary Fig. 6f). We next used RhoB-2A, a mutant with Thr173 and Thr175 mutated to alanines, to perform the in vitro phosphorylation assay. As expected, point mutations of Thr173 and Thr175 to alanines strongly attenuated

the Chk1-mediated phosphorylation of RhoB (Fig. 6e), confirming that the Chk1-mediated RhoB phosphorylation is at Thr173 and Thr175 of RhoB.

To investigate the function of Chk1-mediated phosphorylation of RhoB, we compared the sumoylation and subcellular localization of wild-type RhoB with nonphospho-mimicking mutant RhoB-2A (T173A and T175A) and phospho-mimicking mutant RhoB-2E (T173E and T175E). Remarkably, UV or MMS treatment drastically enhanced interaction of PIAS1 with wild-type RhoB. The RhoB-2E mutant, however, showed a strong interaction with PIAS1 even at basal state, whereas the interaction between RhoB-2A mutant and PIAS1 remained weak even after UV or MMS treatment (Fig. 6f; Supplementary Fig. 6g), indicating that Chk1-mediated phosphorylation of RhoB is important for its binding to PIAS1. Accordingly, UV or MMS treatment significantly increased sumoylation of wild-type RhoB but not RhoB-2A, and RhoB-2E was markedly sumoylated even without UV or MMS treatment (Fig. 6g). In line with this, RhoB-2E colocalized with LAMP1 at basal state, and the 2A mutation abolished the colocalization of RhoB with LAMP1 induced by UV or MMS treatment (Fig. 6h, Supplementary Fig. 6h). It is noticeable that RhoB-2E was mainly in the cytosol at basal state and RhoB-2A retained on plasma membrane even after UV or MMS treatment, suggesting that the phosphorylation of RhoB is required for its dissociation from plasma membrane. Moreover, the 4KR mutation blocked the translocation of RhoB-2E to lysosomes (Supplementary Fig. 6i, j), further confirming that sumoylation is critical for lysosomal translocation of RhoB but not dissociation from plasma membrane.

**Sumoylated RhoB recruits TSC complex to lysosomes**. We then sought to investigate how RhoB promotes autophagy in response to UV or MMS treatment. As shown in Fig. 3, RhoB is required for the formation of both autophagosomes and autolysosomes, suggesting RhoB might be involved in initiating autophagy. We therefore determined the activity of ULK1 by examining its phosphorylation level. Interestingly, we found that knockout of *RhoB* drastically impeded UV-induced or MMS-induced downregulation of phosphorylation of ULK1 and ribosomal protein S6 kinase (S6K), a classical substrate of mTORC1 and its phosphorylation levels are usually used to reflect mTORC1 activity[7] (Fig. 7a), suggesting a critical role of RhoB in inhibiting mTORC1 activity in the context of UV or MMS-induced DNA damage. In contrast, knockout of *RhoB* had no effect on CPT or DOX treatment-induced downregulation of phosphorylation of ULK1 and S6K (Supplementary Fig. 7a), in line with that CPT and DOX do not promote RhoB sumoylation. Indeed, reintroduction of wild-type RhoB but not RhoB-4KR into the *RhoB*[−/−] cells restored the downregulation of phospho-ULK1 and phospho-S6K caused by UV or MMS treatment (Fig. 7b), indicating that sumoylation is required for RhoB-mediated inhibition of mTORC1. This was further confirmed by knocking down PIAS1 (Supplementary Fig. 7b).

It is well known that mTORC1 is directly activated by GTP-bound Rheb, and that TSC complex is the only known GAP for Rheb; therefore, inhibition of mTORC1 by TSC complex is through inactivating Rheb[7]. Recent studies have shown that recruitment of TSC complex to lysosomes is a general mechanism to inactivate mTORC1 in response to various cellular stresses[9]. Indeed, knockout of *TSC2* dramatically blocked UV or MMS treatment-induced autophagy (Supplementary Fig. 7c–f). We found that colocalization of endogenous RhoB and TSC2 was dramatically increased after UV or MMS treatment (Fig. 7c; Supplementary Fig. 7g). We therefore investigated whether RhoB is required for the translocation of TSC complex to lysosomes. As

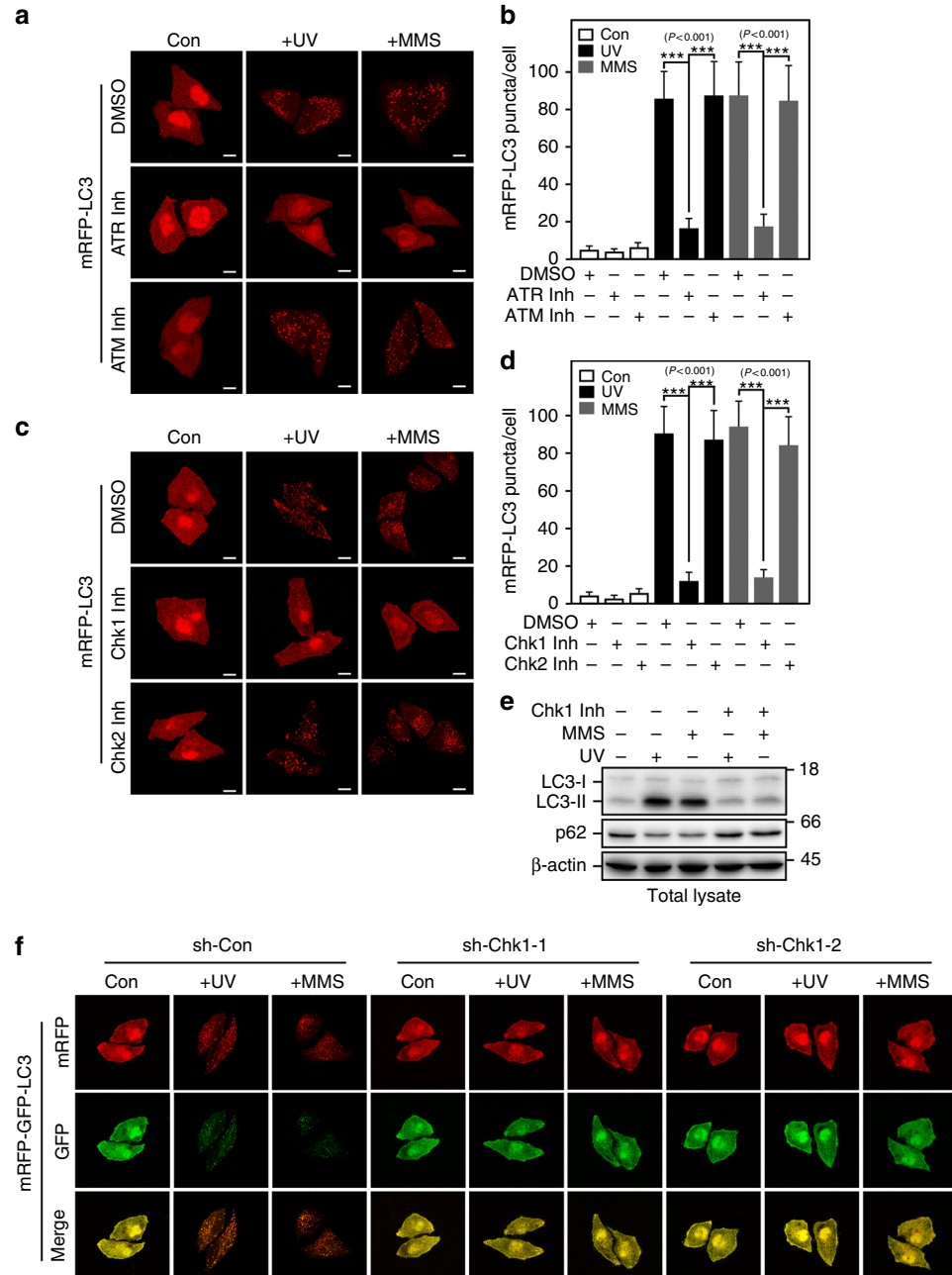

**Fig. 5** ATR/Chk1 activity is required for UV or MMS-induced autophagy. **a** and **b** ATR but not ATM activity is required for UV or MMS-induced LC3 aggregation. HeLa cells with stable expression of mRFP-LC3 pretreated 0.5 h with or without ATM inhibitor CP-466722 (10 μM) or ATR inhibitor VE-821 (1 μM) were subjected to fluorescence microscopy 4 h after UV (80 Jm$^{-2}$) or 6 h after MMS (0.5 mM) treatment (**a**). The numbers of red puncta per cell were quantified using Imaris x64 image analysis software. Five random areas were counted for each experiment and data are presented as mean ± SD of three individual experiments. One-way ANOVA ($F_{(8,126)} = 165.07$) followed by LSD post hoc test for multiple comparisons (**b**). **c** and **d** Chk1 but not Chk2 activity is required for UV or MMS-induced autophagy. HeLa cells with stable expression of mRFP-LC3 pretreated 0.5 h with or without Chk1 inhibitor (0.2 μM) or Chk2 inhibitor-II (4 μM) were treated and subjected to fluorescence microscopy as in panel **a**. Quantification of red puncta per cell was as in panel **b**. One-way ANOVA ($F_{(8,126)} = 229.17$) followed by LSD post hoc test for multiple comparisons. **e** Treatment with Chk1 inhibitor attenuates UV or MMS-induced augment of LC3-II. HeLa cells pretreated 0.5 h with or without Chk1 inhibitor (0.2 μM) were subjected to immunoblotting assay 4 h after treated with UV (80 Jm$^{-2}$) or 6 h after treated with MMS (0.5 mM). **f** Knockdown of Chk1 decreases the autophagic flux promoted by UV or MMS treatment. HeLa cells with stable expression of mRFP-GFP-LC3 and control shRNA (sh-Con) or shRNA against Chk1 (sh-Chk1-1 or sh-Chk1-2) were subjected to fluorescence microscopy 8 h after treated with UV (80 Jm$^{-2}$) or 10 h after treated with MMS (0.5 mM). Scale bar, 10 μm

expected, TSC2 was translocated to lysosomes upon UV or MMS treatment; however, knockout of *RhoB* significantly hindered this translocation (Fig. 7d; Supplementary Fig. 7h). Reintroduction of wild-type RhoB but not RhoB-4KR into the *RhoB*$^{-/-}$ cells

restored the lysosomal translocation of TSC2 in response to UV or MMS treatment (Fig. 7e; Supplementary Fig. 7i). Consistently, knockdown of PIAS1 also remarkably blocked the translocation of TSC2 to lysosomes after UV or MMS treatment

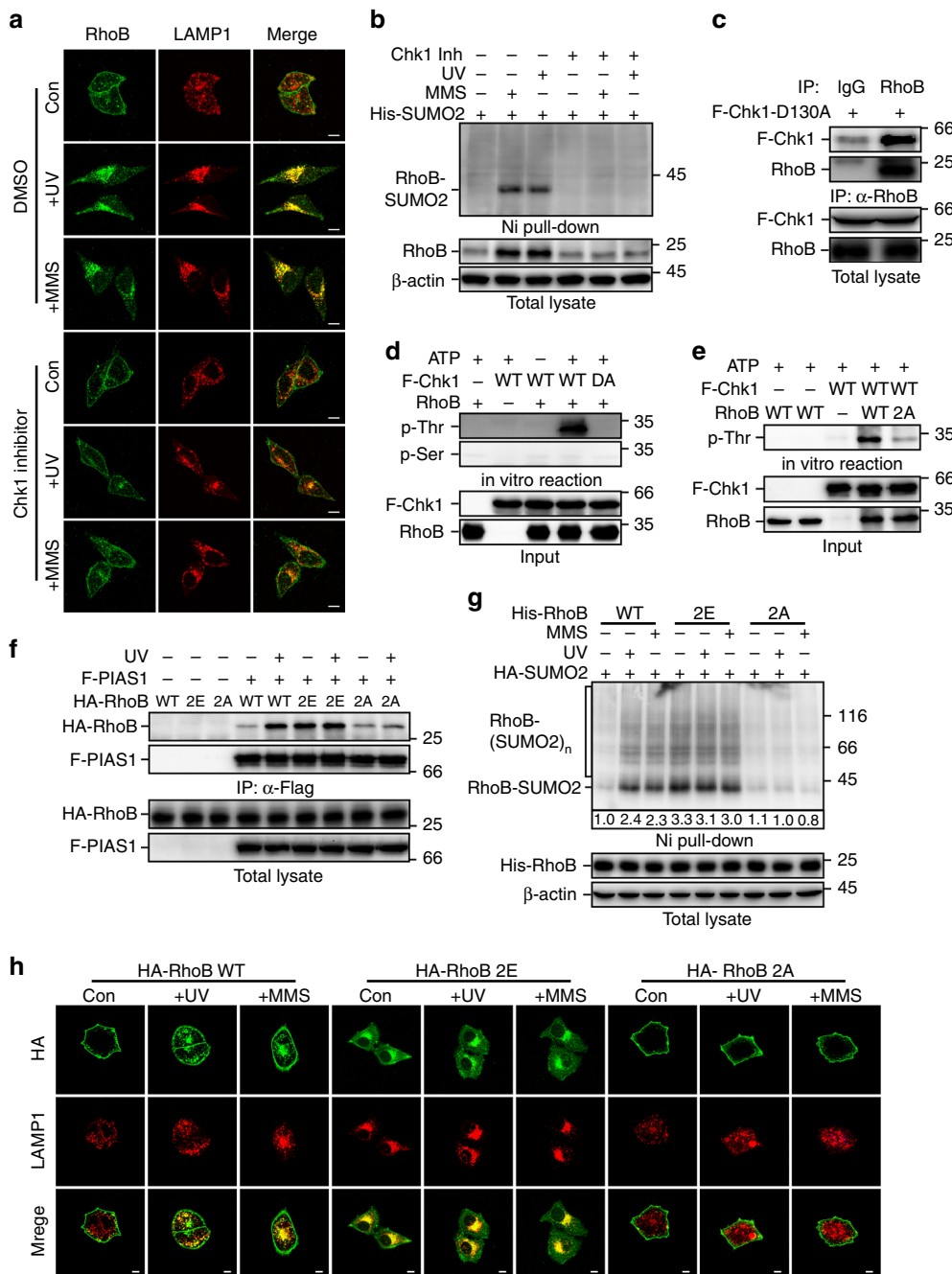

**Fig. 6** Chk1 phosphorylates RhoB to promote its sumoylation and translocation to lysosomes. **a** Chk1 activity is required for translocation of RhoB to lysosomes. HeLa cells pretreated 0.5 h with or without Chk1 inhibitor (0.2 μM) were subjected to immunofluorescence assay 2 h after UV (80 Jm$^{-2}$) or 4 h after MMS (0.5 mM) treatment to examine the localization of endogenous RhoB and LAMP1. Scale bar, 10 μm. **b** Chk1 activity is essential for UV or MMS-induced sumoylation of endogenous RhoB. HeLa cells with expression of His-SUMO2 were treated as in panel **a** and then subjected to sumoylation assay. SUMO2 conjugation of RhoB was detected by immunoblotting with RhoB antibody. **c** Chk1 interacts with endogenous RhoB. HeLa cells transduced with Flag-tagged Chk1-D130A mutant (F-Chk1-D130A) were subjected to anti-RhoB IP followed by immunoblotting with anti-Chk1 to detect associated F-Chk1-D130A. **d** Chk1 phosphorylates RhoB at its threonine residues. In vitro kinase assay was carried out as described in *Methods*. Phosphorylated RhoB was detected by immunoblotting using phospho-threonine or phospho-serine antibodies. **e** Chk1 phosphorylates RhoB at Thr173 and Thr175. RhoB WT or T173,175A mutant (2A) purified from bacteria were used to perform in vitro kinase assay. **f** Phosphorylation of RhoB by Chk1 promotes its binding to PIAS1. HEK293T cells with expression of indicated combination of Flag-tagged PIAS1 (F-PIAS1) and HA-tagged wild-type (WT), T173,175E (2E), or T173,175 A (2A) RhoB were subjected to coimmunoprecipitation assay 1 h after treated with or without UV (80 Jm$^{-2}$) to detect associated RhoB. **g** Phosphorylation of RhoB by Chk1 is required for UV or MMS-induced sumoylation. HEK293T cells with expression of indicated combination of HA-tagged SUMO2 (HA-SUMO2) and His-tagged RhoB (WT, 2E, or 2A) were subjected to sumoylation assay 1 h after treated with UV (80 Jm$^{-2}$) or 2 h after treated with MMS (0.5 mM) to detect the SUMO2 conjugation of RhoB. **h** Chk1-mediated phosphorylation is essential for UV or MMS-induced lysosomal translocation of RhoB. U2OS cells with expression of HA-tagged RhoB (WT, 2E, or 2A) were subjected to immunofluorescence assay 4 h after treated with UV (80 Jm$^{-2}$) or MMS (0.5 mM). Scale bar, 10 μm

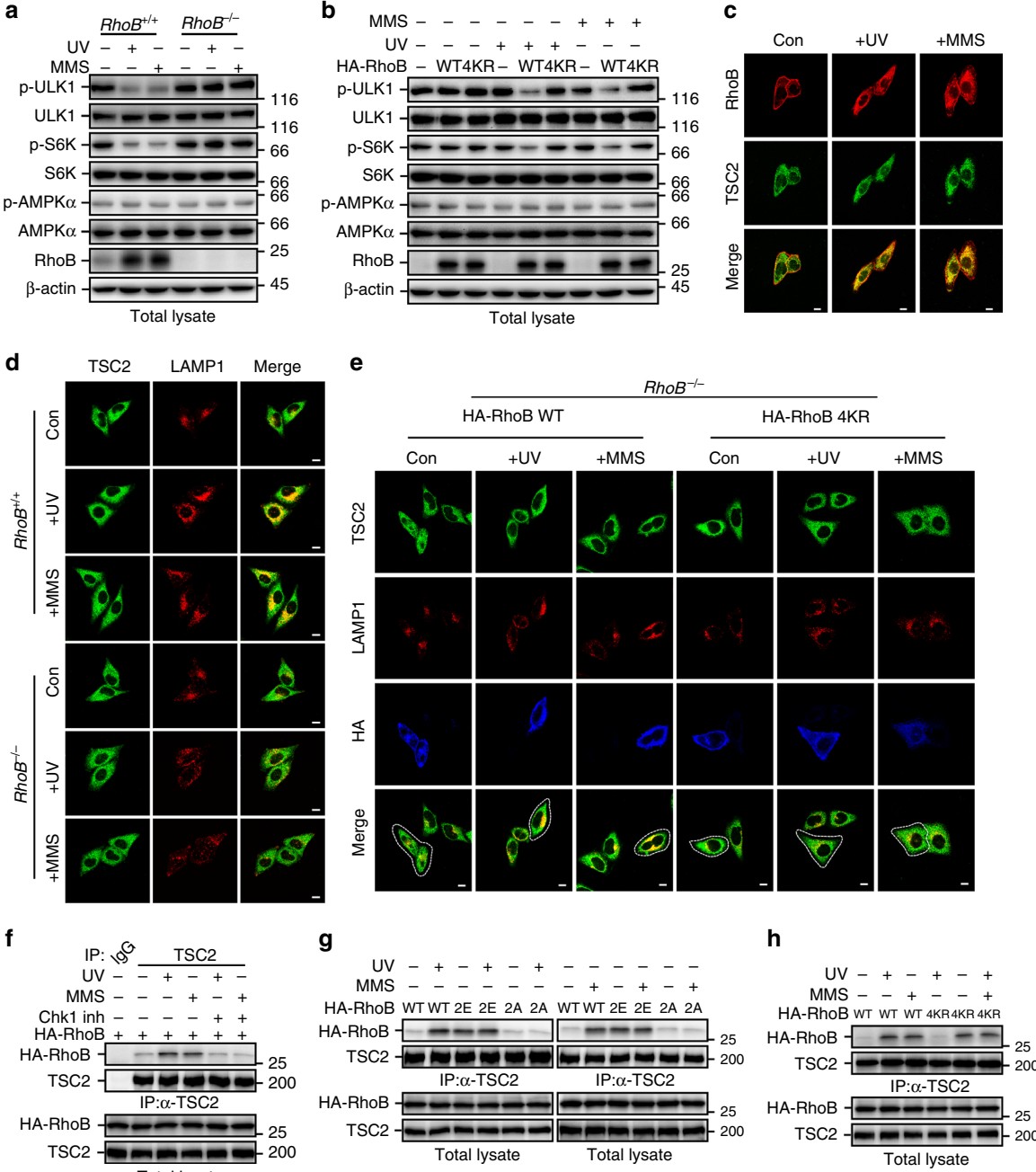

**Fig. 7** Phosphorylated and sumoylated RhoB recruits TSC2 to lysosomes to inhibit mTORC1 activity. **a** Knockout of *RhoB* prevents UV or MMS-induced downregulation of phosphorylated ULK1 and S6K. *RhoB*$^{+/+}$ or *RhoB*$^{-/-}$ cells were subjected to immunoblotting assay 2 h after UV (80 Jm$^{-2}$) or 4 h after MMS (0.5 mM). **b** Sumoylation of RhoB is required for UV or MMS-induced downregulation of phosphorylated ULK1 and S6K. *RhoB*$^{-/-}$ cells transduced with HA-tagged RhoB (WT or 4KR) were treated and subjected to immunoblotting assay as in panel **a**. **c** UV or MMS treatment induces colocalization of endogenous RhoB and TSC2. HeLa cells were treated as in panel **a** and subjected to immunofluorescence assay to examine the localization of endogenous RhoB and TSC2. Scale bar, 10 μm. **d** Knockout of *RhoB* attenuates translocation of TSC2 to lysosomes after UV or MMS treatment. *RhoB*$^{+/+}$ or *RhoB*$^{-/-}$ cells were treated as in panel **a** and subjected to immunofluorescence to examine the colocalization of endogenous TSC2 and LAMP1. **e** Sumoylation of RhoB is critical for UV or MMS-induced lysosomal translocation of TSC2. *RhoB*$^{-/-}$ cells transduced with HA-tagged WT or 4KR RhoB were treated as in panel **a** and subjected to immunofluorescence assay. The dot lines outline cells with expression of RhoB WT or 4KR. Scale bar, 10 μm. **f** UV or MMS treatment enhances interaction between RhoB and endogenous TSC2. HeLa cells were treated as in panel **a** and subjected to anti-TSC2 immunoprecipitation followed by immunoblotting to detect associated RhoB. **g** Phosphorylation of RhoB is required for its interaction with TSC2 induced by UV or MMS treatment. HeLa cells transduced with HA-tagged RhoB (WT, 2E, or 2A) were treated, subjected to anti-TSC2 immunoprecipitation and immunoblotted as in panel **g**. **h** Sumoylation of RhoB is not responsible for its binding to TSC2. HeLa cells transduced with HA-tagged RhoB (WT or 4KR) were treated, subjected to anti-TSC2 immunoprecipitation and immunoblotted as in panel **g**

(Supplementary Fig. 7j, k), indicating a pivotal role of sumoylation in RhoB-mediated translocation of TSC complex to lysosomes.

Next, we found that the interaction between RhoB and endogenous TSC2 was significantly enhanced by UV or MMS treatment, and this interaction was significantly blocked by cotreatment of Chk1 inhibitor (Fig. 7f). Accordingly, Chk1 inhibitor dramatically blocked UV or MMS-induced decrease in phosphorylated ULK1 and S6K (Supplementary Fig. 7l). Meanwhile, RhoB-2E strongly interacted with TSC2 even at basal state, whereas the interaction between TSC2 and RhoB-2A retained at low level even after UV or MMS treatment (Fig. 7g), indicating that phosphorylation of RhoB is required for its interaction with TSC2. Interestingly, wild-type RhoB and RhoB-4KR showed comparable binding affinity to TSC2 at basal state and after UV or MMS treatment (Fig. 7h), indicating that the sumoylation of RhoB does not affect its binding to TSC2. Reintroduction of RhoB-2E, the phospho-mimicking mutant that will be sumoylated even at basal state (Fig. 6g), into $RhoB^{-/-}$ cells significantly promotes translocation of TSC2 to lysosomes, whereas reintroduction of RhoB-2E-4KR did not rescue the translocation (Supplementary Fig. 7m, n), indicating that phosphorylation and sumoylation of RhoB are both required to translocate TSC complex to lysosomes. Hence, our study demonstrated that activation of ATR/Chk1 signaling upon DNA damage triggers a Chk1-mediated phosphorylation of RhoB, which promotes its binding to TSC2 and sumoylation. The sumoylated RhoB will then translocate TSC complex to lysosomes to deactivate mTORC1, thereby initiating autophagy.

## Discussion

It is well known that ATM/Chk2 and ATR/Chk1 signaling cascades are two distinct pathways in response to different types of DNA lesions. ATM-Chk2 pathway is principally activated by double-strand DNA breaks, whereas ATR/Chk1 is usually activated by single-strand DNA lesions[14,15]. Previous studies suggested that reactive oxygen species and reactive nitrogen species can initiate autophagy through ATM-mediated activation of AMPK, which in turn inhibits mTORC1 by activating TSC2[16,40]. However, how ATR is involved in regulating autophagy is not clear yet. In this study, we identified a mechanism for ATR/Chk1 signaling-induced autophagy, in which activation of ATR/Chk1 by DNA damage promotes RhoB phosphorylation and subsequent sumoylation, leading to the translocation of TSC complex to lysosomes to inhibit mTORC1 activity, thereby initiating autophagy. The phosphorylation of RhoB is mainly required for its dissociation from the plasma membrane and further interaction with PIAS1 and TSC2, whereas the sumoylation of RhoB is indispensable for its capability of translocating TSC2 to lysosomes. It is well known that sumoylation plays an important role in regulating protein trafficking[41,42]; however, how phosphorylation results in dissociation of RhoB from plasma membrane is unclear. The plasma membrane localization of RhoB is dependent on the post-translational lipid modifications at its C-terminal[43], it is possible that phosphorylation of RhoB affects the lipid modifications by a yet to be identified mechanism. Our previous study showed that downregulation of Smurf1 and the resultant upregulation of RhoB is a specific event downstream of ATR/Chk1 signaling pathway under DNA damage stress[36]. Therefore, the ATR/Chk1 signaling regulates RhoB at two different levels. On one hand, ATR/Chk1 stabilizes RhoB through inactivation of Smurf1; on the other hand, it phosphorylates RhoB and promotes RhoB sumoylation for further translocation of TSC complex to lysosomes to inhibit mTORC1 and initiate autophagy.

Recent studies showed that depletion of growth factors or amino acids results in a translocation of TSC1/2 complex to lysosomal surface to inhibit mTORC1 activity[8,10], and lysosomal recruitment of TSC2 also happens upon different cellular stresses including hyperosmotic stress, pH stress, hypoxic stress, etc.[9]. It was suggested that although lysosomal relocalization of TSC complex could be a general response to different stresses, the underlying molecular mechanism for regulating the localization of TSC complex may vary from stress to stress. However, the molecular mechanisms by which TSC complex is recruited to lysosomes in response to these stresses remain largely unknown[9]. In this study, we found that DNA damage caused by UV or MMS treatment also recruits TSC2 to lysosomes. More importantly, we revealed a unique mechanism of sumoylated RhoB-mediated lysosomal translocation of TSC2 in response to DNA damage. The sumoylation of RhoB is only promoted by UV or MMS-induced DNA damage, but not CPT or DOX-induced double-strand DNA damage (Fig. 2a), indicating that different stresses indeed work through distinct pathways to regulate the subcellular localization of TSC complex. Further study to elucidate the molecular mechanisms for spatial control of TSC complex localization under different stresses will be of great interest and importance to a deeper understanding of how mTORC1 activity is manipulated in response to various stimuli.

Autophagy primarily acts as a protective mechanism for cell survival in response to various stresses including nutrient starvation, growth factor deprivation, reactive oxygen species, hypoxia, damaged organelles, protein aggregation, and DNA damage[44]. Nevertheless, increasing evidence indicates that excessive autophagy above certain threshold eventually leads to programmed cell death[12,45]. However, how the extent of autophagy is regulated under these stresses is still elusive. RhoB is recognized as a tumor suppressor due to its role in regulating cell cycle and apoptosis[26]. Our previous study showed that accumulation of RhoB by ATR/Chk1-promoted self-degradation of Smurf1, a major E3 ubiquitin ligase responsible for RhoB turnover, is required for UV or MMS-triggered apoptosis[36]. Interestingly, we observed that the activity of RhoB for enhancing cell death after UV or MMS treatment is largely dependent on its capability of promoting autophagy. Ablating sumoylation of RhoB blocked translocation of RhoB to lysosomes and simultaneously prohibited RhoB-elevated cell death after DNA damage. Meanwhile, double knockout of *ULK1/2* remarkably abolished the RhoB-promoted cell death, suggesting that RhoB-mediated apoptosis might be through an autophagy-dependent pathway. In fact, recent studies indicate that autophagy may either directly mediate cell death or facilitate activation of apoptosis[46]. Hence, further scrutinize the relationship between the RhoB-mediated autophagy and RhoB-mediated apoptosis in response to ATR/Chk1 signaling may bring new insights into understanding the molecular mechanisms underlying the interplay between autophagy and apoptosis.

## Methods

**DNA constructs**. Human RhoB constructs (wild-type and K6,7R) have been previously described[36]. The cDNAs for UBC9, SUMO1/2, PIAS1/2/3/4, SAE1, UBA2, FIS1, LC3B, and TSC2 were generous gifts from Dr. J.Han. RhoB mutants (K135R, K194R, 4KR, T173, 175A, T173, 175E) were generated by PCR-based site-directed mutagenesis. Cloning for protein expression in mammalian cells was carried out using a modified pCMV5 vector for transfection, pBOBI and pCDH-EF1-MCS-IRES-puro vectors for lentivirus infection. pET28a-E1/E2/S2 plasmid was generated according to previous report except pET28a vector was used instead of pT-Trx[47]. The lentivirus-based vector pLL3.7 was used for expression of shRNA. The sequences for ATR shRNA, Chk1 shRNA-1 and -2, and control shRNA have been described in previous report[36]. The sequences used for PIAS1 shRNA-1 and -2 are 5′-GCTCCATATGAACACCTTA-3′ and 5′-AGATGTTTCTTGATCAGTT-3′, respectively.

**Antibodies and chemical reagents**. Mouse anti-actin (1:2000, sc-47778), anti-Myc (1:2000, sc-40), anti-RhoB (1:1000, sc-8048), anti-GST (1:2000, sc-138), anti-UBC9 (1:2000, sc-271057), anti-LAMP1 (1:200, sc-20011), anti-tuberin (1:1000, sc-271314), rabbit anti-Chk1 (1:1000, sc-7898), and rabbit anti-RhoB (1:1000, sc-180) antibodies were purchased from Santa Cruz Biotechnology (Santa Cruz, CA, USA); mouse anti-HSP60 (1:200, H3524) and anti-FLAG (M2) (1:2000, F1804) antibodies were purchased from Sigma-Aldrich (St Louis, MO, USA); rabbit anti-TSC2 (1:1000, #4308), anti-ULK1 (1:1000, #8054), anti-AMPKα (1:1000, #2532), anti-PIAS1 (1:1000, #3550), anti-p70S6 kinase (1:1000, #2708), anti-phospho-p70S6 kinase (1:1000, #9205), anti-phospho-ULK1 (1:1000, Ser757#14202), anti-phospho-AMPKα (1:1000, Thr172#2531), anti-HSP60 (1:1000, #4870), anti-phospho-Threonine (1:1000, #9381), anti-Chk2 (1:1000, #2662), anti-Phospho-Chk1 (1:1000, ser345, #2348), and anti-Phospho-Chk2 (1:1000, Thr68, #2661) antibodies were purchased from Cell Signaling Technology; mouse anti-SQSTM1/P62 (1:1000, ab56416) antibody was purchased from Abcam; rat anti-HA (1:2000, #11867431001) monoclonal antibody was purchased from Roche (Mannheim, Germany); rabbit anti-LC3B/MAP1LC3B (1:1000, NB100-2220) was purchased from Novus; mouse anti-phosphoserine (1:1000, #05-1000), mouse anti-phospho-Histone H2A.X (Ser139)(1:200, #05-636) were purchased from Millipore; rabbit anti-RhoB (1:1000, 14326-1-AP) and rabbit anti-ATR (1:1000, 19787-1-AP) were purchased from Proteintech; inhibitors for Chk1 (#681637) and Chk2 (#220485) were purchased from Calbiochem; inhibitors for ATM CP-466722(#11002) and ATR VE-821(#14731) were purchased from MedChemExpress; chloroquine, methyl methanesulphonate, guanidine hydrochloride, and urea were purchased from Sigma-Aldrich (St Louis, MO, USA); Doxorubicin (HY-15142) and Campathecin (HY-16560) were purchased from MedChemExpress. All the original blots with size markers are available in the Supplementary Information.

**Cell culture, transfection, and lentivirus infection.** Human embryonic kidney HEK293T, human cervical cancer HeLa, and human osteosarcoma U2OS were obtained from ATCC. $ULK1/2^{+/+}$ and $ULK1/2^{-/-}$ MEF cells were a kind gift from Dr. S.C. Lin; $TSC2^{+/+}$; and $TSC2^{-/-}$ MEF cells were a kind gift from Dr. Q. Wu. The cells were cultured in high-glucose Dulbecco's modified Eagle's medium (DMEM) supplemented with 10% (v/v) fetal bovine serum (FBS) (Thermo) and 100 units/ml streptomycin and penicillin (Millipore) at 37°C in a humidified 5% $CO_2$ incubator. The cell lines were routinely tested and found negative for mycoplasma. Transient transfection of HEK293T cells was performed as previously described[48]. Infection of HeLa, U2OS, and MEF cells were carried out using recombinant lentivirus generated through the ViraPower Lentiviral Expression System (Invitrogen).

**Generation of *RhoB* knockout cell line.** HeLa cells were used to generate *RhoB* knockout cell line using pX330 CRISPR/Cas9[49] vector. The gene-specific region of the gRNA sequences were designed by the CRISPR design tool from Zhang lab (http://crispr.mit.edu/) and the two gRNA sequences RhoB-1 and -2 are 5′-CTTGCGCCAGGACTTGGCGT-3′ and 5′-GAGCAGCGCGGGCGAGACGCA-3′, respectively. The pX330 empty vector was used as control. Single clones were picked up and the efficiency of *RhoB* knockout was assessed by western blot.

**Immunoprecipitation, immunoblotting, and GST pull-down assays.** Immunoprecipitation, immunoblotting, GST pull-down assays were performed as previously described[48]. Briefly, cells were lysed on ice with lysis buffer TNTE 0.5% (50 mM Tris-HCl, pH 7.5, 150 mM NaCl, 1 mM EDTA, and 0.5% Triton X-100, containing 10 μg/ml pepstatin A, 10 μg/ml leupeptin, and 1 mM PMSF) and then applied to immunoprecipitation or immunoblotting assays with appropriate antibodies. For GST pull-down assay, bacterially expressed GST-RhoB, GST-Ubc9, GST-PIAS1, and GST-RBD were purified using glutathione sepharose beads in TNTE 0.5% buffer. Free Ubc9 and PIAS1 proteins were obtained by tobacco etch virus (TEV) protease cleavage. Uncropped scans have been included in the Supplementary file (Supplementary Fig. 8).

**In vivo and in vitro sumoylation assays.** The in vivo sumoylation assay was carried out as previously described[50]. Briefly, cells were lysed with the lysis buffer (6 M guanidine-HCl, 0.1 M $Na_2HPO_4/NaH_2PO_4$, 0.01 M Tris/HCl, pH 8.0, 5 mM imidazole, 20 mM N-ethylmaleimide, and 10 mM β-mercaptoethanol). The lysates were sonicated and centrifuged at 135,000 g at room temperature for 15 min to collect supernatant. The supernatant was incubated with $Ni^{2+}$-NTA-agarose beads at 4°C overnight. The beads was washed thoroughly and boiled with loading buffer before applied to immunoblotting assay.

For in vitro sumoylation assay, the plasmids pET28a-E1/E2/S2 and pGEX-RhoB (WT or 4KR) were simultaneously introduced into E. coli BL21 with selection of both Ampicilin and Kanamycin based on previous report[47]. The bacterial lysate was subjected to GST pull-down followed by western blot to detect the sumoylated RhoB.

**In vitro kinase assay.** HEK293T cells transfected with triple Flag-tagged wild-type Chk1 or its dominant negative form Chk1-D130A were pretreated with UV (80 J/m²) and subjected to immunoprecipitation with anti-Flag antibody 2 h later, followed by washing twice with 0.5% TNTE and twice with kinase reaction buffer (20 mM Tris-HCl pH 7.5, 10 mM $MgCl_2$, 1 mM dithiothreitol, 25 μM ATP). Wild-type or mutant

RhoB purified from bacteria was incubated with Chk1 in a total volume of 20 μl of reaction buffer at 37 °C for 30 min before subjected to immunoblotting assay.

**Immunofluorescence assay.** Cells grown on glass coverslips were washed three times with phosphate-buffered saline (PBS), fixed with 4% paraformaldehyde and permeabilized with methanol at −20 °C. The cells were then stained using appropriate primary and fluorescently conjugated secondary antibodies. Secondary antibodies used for this assay were Alexa Fluor 405 or 555-conjugated donkey anti-mouse, Alexa Fluor 488-conjugated donkey anti-rabbit, and Alexa Fluor 488 or 647-conjugated donkey anti-rat secondary antibodies (Invitrogen). Texas Red conjugated Phalloidin was used for F-actin staining. Images were obtained using a ZEISS LSM 780 confocal microscope with ZEN 2010 software (Carl Zeiss GmbH, Jena, Germany) or a Leica TCS SP8 confocal microscope with LAS AF software (Leica, Germany).

**Transmission electron microscopy.** HeLa cells were fixed with 2.5% glutaraldehyde in 0.1 M PBS (pH 7.4) at 4 °C for 2.5 h, washed three times with 0.1 M PBS and post-fixed in 1% $OsO_4$ at 4 °C for 2 h. The samples were subsequently dehydrated in an ethanol gradient (30% (v/v) ethanol (15 min), 50% (v/v) ethanol (15 min), 70% (v/v) ethanol (15 min), 90% (v/v) ethanol (15 min), and 100% ethanol (2 × 20 min)) and embedded in Spurr's resin. Ultrathin (60 nm) sections were then collected on a copper grid, stained with either uranyl acetate or lead citrate and examined by a JEM2100HC transmission electron microscope (Hitachi).

**UV irradiation and MMS treatment.** UV irradiation and MMS treatment were performed as previously described[36]. Briefly, cells were exposed to UV (80 Jm⁻²) in Spectrolinker XL-1000 (Spectronics) and then incubated in the original medium for determined time. For MMS treatment, cells were treated with 0.5 mM MMS for determined time before subjected to further analysis.

**Apoptosis assay.** Apoptosis assay was performed as previously described[36]. Briefly, cells treated with UV or MMS were collected and washed in PBS before fixed with 70% ethanol overnight. Fixed cells were washed and incubated with 100 U ml⁻¹ RNaseA in PBS at 37°C for 30 min, and then stained with 50 mg ml⁻¹ propidium iodide in PBS for another 30 min at 37°C in the dark. The percentages of apoptotic cells were determined by calculating sub-G1 proportion of cells using a Beckman Coulter FC500 flow cytometer (Beckman Coulter, Indianapolis, IN, USA).

**Microscopic data analysis.** Punctum formation was assessed by using Imaris x64 image analysis software (version 7.2.3, Bitplane). In brief, distinct puncta were identified using the spot-create function. All images were batch analyzed using the same threshold. In autophagy flux assay using mRFP-GFP-LC3, puncta with both red and green fluorescence that appeared yellow in merged images were counted as autophagosomes; and puncta appeared red in merged images were counted as autolysosomes.

**Statistical analysis.** One-way ANOVA with LSD post hoc test was used to compare values among different experimental groups using the SPSS Statistics 18.0 program. $p < 0.05$ was considered a statistically significant change. *$p < 0.05$; **$p < 0.01$; ***$p < 0.001$; NS, not significant. All the values were presented as mean ± SD of at least triplicate experiments.

## Data availability

All data supporting the findings of this study are available within the article and its Supplementary Information files or from the corresponding author upon request.

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

## Acknowledgements

This work was supported by the National Natural Science Foundation of China (U1605222, 81472459, 31671223), the National Key Research and Development Project of China (2016YFC1302400, 2016YFA0502003), the Fundamental Research Funds for the Central Universities (20720140550, 20720160070), the National Science Foundation for Fostering Talents in Basic Research of the National Natural Science Foundation of China (J1310027), the Project 111 sponsored by the State Bureau of Foreign Experts and Ministry of Education (B12001), the National Natural Science Foundation of China (31601132) to T.Z., the National Natural Science Foundation of China (81402290) to Q.L., and the National Natural Science Foundation of China (U1405223) to X.D.

## Author contributions

M.L., T.Z. and X.Z. conducted the experiments and analyzed the data. C.L., Z.W. and H.X. performed molecular biology experiments. L.Y. carried out electron microscope and C.X. performed mass spectrometry. Q.L., L.X. and H.-L.C. analyzed the data. D.Z. and X.D. contributed reagents and materials. T.-J.Z. and H.-R.W. designed the experiments and wrote the paper.

## Additional information

**Competing interests:** The authors declare no competing interests.

