## [Peer Review File · Nature Communications]

Reviewers' comments:

Reviewer #1 (Remarks to the Author):

The manuscript by the Wang and Zhao groups defines the pathway transducing the signals from DNA damage to the growth suppression and the activation of autophagy. They propose that the lysosomal localization of TSC2 and mTOR is a key step in this regulation. The small GTPase RhoB, whose phosphorylation and sumoylation is induced by the DNA damage response (DDR), links DNA damage to the TSC2/mTOR cassette.

Overall, the data are convincing and well presented. The experimental approach exclusively uses cultured cells and signal transduction techniques. The physiological *in vivo* evidence is not addressed in this study. However, the proposed molecular mechanism may be of general interest.

The following points should strengthen the conclusion on the mTOR pathway and cellular outputs:

- 1) The antibodies used for immunofluorescence localization of endogenous proteins (RhoB and TSC2) should be validated in knockout or knockdown cells to demonstrate their specificity. This is particularly important for TSC2, whose localization in the lysosomes is not convincing.
- 2) The specificity for UV and MMS in inducing RhoB, as opposed to cisplatin and dox (Fig. 2), should be further corroborated. DNA damage and DDR should be shown for cisplatin and dox. Similarly, mTOR suppression and autophagy activation should be also assessed for cisplatin and dox.
- 3) The TSC2 localization and translocation to lysosomes is not convincing (Fig. 7B and 7C). Serum starvation and stimulation should be used as a control. Specificity of the antibody should be provided (see above). The lower co-localization signal of TSC2 and Lamp1 appears mainly to be due to decreased Lamp1 signal rather than TSC2 localization in RhoB mutant cells (Fig. 7B). Does RhoB deletion affect lysosome biogenesis? What is the phosphorylation and activity of TFEB in these cells?
- 4) The mTOR activity after DNA damage in RhoB mutant cells and control should also be assessed in serum and amino acid starvation conditions (Fig. 7A), to show whether RhoB deletion provides a more general activation of mTOR.
- 5) TSC knockout/knockdown cells should be used to demonstrate that the effects of DNA damage and RhoB on autophagy are really dependent on TSC2.
- 6) The physiological outcome of the DDR-RhoB dependent autophagy induction on apoptosis/survival/senescence/proliferation/growth would be a nice addition to the study.

Minor points:

- 7) The quality of EM pictures (Fig. 3F and supplementary) is rather poor. Resolution should be improved. Arrows should be used to indicate the organelles. Mitochondria and mitophagy are not clearly visible.
- 8) The knockdown efficiency for the sh-Chk1 experiments should be shown.

Reviewer #2 (Remarks to the Author):

In this manuscript, Liu and colleagues have shown that DNA damaged caused by UV or MMS phosphorylates RhoB by Chk1 that promotes the sumoylation of RhoB, and both these post-translational modifications of RhoB leads to its interaction with TSC2, translocation of the RhoB/TSC2 complex to the lysosomes, inhibition of mTORC1 and induction of autophagy. This is an interesting study which has been undertaken very well, the manuscript is clearly written, data quality is excellent, and the overall results support the conclusions made. The work is convincing and shows novel insights into the regulation of autophagy by DNA damage. The comments below will help to further improve the manuscript, particularly the demonstration of this mechanism in disease-relevant scenarios associated with DNA damage like cancer.

1. Can knockdown of PIAS1 abrogate UV/MMS-induced sumoylated RhoB-mediated autophagy?
2. Show Rheb activity in wild-type and knockout RhoB cells after UV/MMS treatment, and the effects of Chk1 inhibitor.
3. Show effects on mTORC1 activity in UV/MMS treated cells with PIAS1 knockdown or Chk1 inhibitor.
4. Show shRNA-mediated knockdown efficacy of ATR, Chk1 and Chk2.
5. Quantify the microscopy images of the key colocalization experiments.
6. In many instances, HeLa and HEK293 cells have been used for experimentation in random order. Describe the choice of cell types and the reason for switching between them for various experiments.
7. To ascertain the disease relevance, can the mechanism proposed here be demonstrated in any pathological conditions such as cancer associated with DNA damage?
8. Discussion section should describe why phosphorylation of RhoB is required for dissociation from plasma membrane, and why sumoylation of RhoB is required for lysosomal translocation?

Response to Reviewers

Reviewer #1 (Remarks to the Author):

The manuscript by the Wang and Zhao groups defines the pathway transducing the signals from DNA damage to the growth suppression and the activation of autophagy. They propose that the lysosomal localization of TSC2 and mTOR is a key step in this regulation. The small GTPase RhoB, whose phosphorylation and sumoylation is induced by the DNA damage response (DDR), links DNA damage to the TSC2/mTOR cassette. Overall, the data are convincing and well presented. The experimental approach exclusively uses cultured cells and signal transduction techniques. The physiological in vivo evidence is not addressed in this study. However, the proposed molecular mechanism may be of general interest. The following points should strengthen the conclusion on the mTOR pathway and cellular outputs:

Q1. The antibodies used for immunofluorescence localization of endogenous proteins (RhoB and TSC2) should be validated in knockout or knockdown cells to demonstrate their specificity. This is particularly important for TSC2, whose localization in the lysosomes is not convincing.

Response: As suggested by the reviewer, we examined the specificity of anti-RhoB and anti-TSC2 antibodies using *RhoB*^{-/-} and *TSC2*^{-/-} cells, respectively. As shown in Fig. for reviewers 1, the anti-RhoB and anti-TSC2 antibodies only stained in wild-type cells but not the knockout cells, indicating that these antibodies are specific for immunostaining RhoB or TSC2.

Q2. The specificity for UV and MMS in inducing RhoB, as opposed to cisplatin and dox (Fig. 2), should be further corroborated. DNA damage and DDR should be shown for cisplatin and dox. Similarly, mTOR suppression and autophagy activation should be also assessed for cisplatin and dox.

Response: We thank for the reviewer's insightful suggestion for further examination of the effects of cisplatin on mTOR suppression. It is likely due to the multiple roles of cisplatin in affecting diverse signaling pathways, especially PI3-K/AKT, MAPK, and ATR pathways (*Oncogene* 2003, 22, 7265–7279) we found cisplatin has a complex role in regulating mTORC1 activity. As shown in Figure for reviewers 2, cisplatin activates mTORC1 at lower doses and inhibits mTORC1 activity at higher doses. Therefore, we used topoisomerase I inhibitor camptothecin (CPT), which also mainly activates Chk2 but not Chk1 as DOX does (revised Supplementary Fig. 2b), to replace cisplatin as a control in our paper (revised Fig. 2a, 2c, 3a; revised Supplementary Fig. 2a, 2b, 2d, 3a, 3c, 3d, 7a). Of note, both DOX and CPT suppressed mTORC1 activity in a dose dependent manner (Fig. for reviewers 2).

Although UV, MMS, CPT, and DOX all strongly induced DNA damage as

represented by γ -H2A.X staining (revised Supplementary Fig. 2a), UV or MMS-induced DDR was mainly through Chk1 pathway, whereas CPT or DOX had much less potency on Chk1 activation (revised Supplementary Fig. 2b). As a consequence, CPT or DOX did not enhance sumoylation of RhoB (revised Fig. 2a), neither did they promote the lysosomal translocation of RhoB (Fig. 2c). Accordingly, knockout of *RhoB* only inhibited UV or MMS-mediated mTOR suppression (revised Fig. 7a) and autophagy (revised Fig. 3a, 3b; revised Supplementary Fig. 3a, 3b), but not CPT or DOX-mediated mTOR suppression (revised Supplementary Fig. 7a) or autophagy (revised Fig. 3a; revised Supplementary Fig. 3a, 3c, 3d).

Q3. The TSC2 localization and translocation to lysosomes is not convincing (Fig. 7B and 7C). Serum starvation and stimulation should be used as a control. Specificity of the antibody should be provided (see above). The lower co-localization signal of TSC2 and Lamp1 appears mainly to be due to decreased Lamp1 signal rather than TSC2 localization in *RhoB* mutant cells (Fig. 7B). Does *RhoB* deletion affect lysosome biogenesis? What is the phosphorylation and activity of TFEB in these cells?

Response: As suggested by the reviewer, we examined the TSC2 and LAMP1 localization in response to amino acids, glucose, or serum starvation. All these treatments showed similar effect on TSC2 and LAMP1 localization with UV or MMS treatment, (previous Fig. 7b, revised Fig. 7d and Fig. for reviewers 3a). In contrast to UV or MMS treatment, knockout of *RhoB* does not affect the lysosomal translocation of TSC2 induced by these treatments.

As shown in Fig. for reviewers 3b, LAMP1 and LAMP2 was not decreased in *RhoB*^{-/-} cells, and treatment of UV or MMS did not affect the protein levels of LAMP1 and LAMP2, indicating the decrease of immunofluorescence signal of LAMP1 in UV or MMS treated *RhoB*^{-/-} cells is most likely due to a blockade of LAMP1 aggregation. Interestingly, protein levels of LAMP1 and LAMP2 were rather increased in *RhoB*^{-/-} cells, suggesting that knockout of *RhoB* might block turnover of LAMP1 and LAMP2.

The knockout of *RhoB* has minimal effect on phosphorylation of TFEB at basal state, indicating *RhoB* deletion itself does not affect lysosome biogenesis under normal conditions. Moreover, TFEB is also a substrate of mTORC1 (*J Cell Sci.* 2016, 129, 2475-2481). TFEB is phosphorylated by mTORC1 at basal state, and UV or MMS treatment drastically caused dephosphorylation of TFEB, likely through inactivating mTORC1. The dephosphorylation of TFEB induced by UV or MMS treatment was significantly blocked in *RhoB*^{-/-} cells, suggesting that inactivation of mTOR was prohibited in *RhoB*^{-/-} cells, which is in good agreement with the phospho-S6K results (Fig. 7a).

Q4. The mTOR activity after DNA damage in RhoB mutant cells and control should also be assessed in serum and amino acid starvation conditions (Fig. 7A), to show whether RhoB deletion provides a more general activation of mTOR.

Response: Following the reviewer's suggestion, we investigated whether RhoB is required for serum, amino acid, or glucose starvation-induced inactivation of mTOR. As shown in Fig. for reviewers 3c, knockout of *RhoB* showed no effect on dephosphorylation of ULK1 or S6K caused by these treatments, indicating that RhoB is not involved in regulation of mTOR activity under these conditions.

Q5. TSC knockout/knockdown cells should be used to demonstrate that the effects of DNA damage and RhoB on autophagy are really dependent on TSC2.

Response: As suggested by the reviewer, we examined the necessity of TSC2 in UV or MMS-induced autophagy. As shown in revised Supplementary Fig. 7c-f, knockout of *TSC2* dramatically blocked UV or MMS treatment-triggered puncta formation of mRFP-LC3, up-regulation of LC3-II, and down-regulation of p62, indicating that TSC2 is indeed required for UV or MMS-induced autophagy.

Q6. The physiological outcome of the DDR-RhoB dependent autophagy induction on apoptosis/survival/senescence/proliferation/growth would be a nice addition to the study.

Response: Following the reviewer's insightful suggestion, we further explored the cellular function of DDR-RhoB dependent autophagy. We found that sumoylation is required for RhoB-promoted cell death, and double *knockout* of *ULK1/2* significantly attenuated this effect, indicating that RhoB-mediated autophagy has an important role in promoting cell death after DNA damage (revised Fig. 4g-i).

Minor points:

Q7. The quality of EM pictures (Fig. 3F and supplementary) is rather poor. Resolution should be improved. Arrows should be used to indicate the organelles. Mitochondria and mitophagy are not clearly visible.

Response: We are sorry that the quality of EM pictures in our original manuscript was poor due to the loss of resolution during generating PDF. We now provided higher resolution pictures in our revised manuscript (revised Fig. 3g; revised Supplementary Fig. 3n), in which clearer images can be seen. Meanwhile, we added arrows to indicate the autophagosomes/autolysosomes with mitochondria.

Q8. The knockdown efficiency for the sh-Chk1 experiments should be shown.

Response: The knockdown efficiency for the sh-Chk1 has been provided in our revised manuscript as the reviewer suggested (revised Supplementary Fig. 5g).

Reviewer #2 (Remarks to the Author):

In this manuscript, Liu and colleagues have shown that DNA damaged caused by UV or MMS phosphorylates RhoB by Chk1 that promotes the sumoylation of RhoB, and both these post-translational modifications of RhoB leads to its interaction with TSC2, translocation of the RhoB/TSC2 complex to the lysosomes, inhibition of mTORC1 and induction of autophagy. This is an interesting study which has been undertaken very well, the manuscript is clearly written, data quality is excellent, and the overall results support the conclusions made. The work is convincing and shows novel insights into the regulation of autophagy by DNA damage. The comments below will help to further improve the manuscript, particularly the demonstration of this mechanism in disease-relevant scenarios associated with DNA damage like cancer.

Q1. Can knockdown of PIAS1 abrogate UV/MMS-induced sumoylated RhoB-mediated autophagy?

Response: According to the reviewer's suggestion, we examined the effect of PIAS1 knockdown on RhoB-mediated autophagy. Indeed, knockdown of PIAS1 significantly blocked UV or MMS-induced autophagy (revised Supplementary Fig. 3g-j), which is in good agreement with our conclusion that sumoylation is necessary for RhoB-induced autophagy.

Q2. Show Rheb activity in wild-type and knockout RhoB cells after UV/MMS treatment, and the effects of Chk1 inhibitor.

Response: Structurally related Ras binding domains (RBDs) are frequently used to extract GTP bound Ras family small GTPases. However, for many Ras family members including Rheb, there are not well defined RBDs can be used to pull down these small GTPases. Although it was claimed that GTP-bound Rheb interacts with FKBP38 (*Science*, 2007, 318, 977-980), contradictory data had been reported by several groups (*J Biol Chem.* 2008, 283, 30482-92; *FEBS Lett.* 2009, 583, 965-70; *J Biol Chem.* 2009, 284, 12783-91). We performed GST-pull down experiments to examine the interaction between GST-FKBP38 and Rheb, but we could not see specific interaction between GTP-bound Rheb and FKBP38.

Because there is no effective GST-RBD pull down assay for Rheb, the only way to measure the levels of its GDP-bound form is by ³²p labeling experiment. According to the literature (*Methods* 2005, 37, 190-196), it needs 5mCi ³²p to label endogenous Rheb for each sample, which can only give relative weak signal as shown in the following figure. Therefore, to examine Rheb activity as the reviewer suggested, we need at least 30 mCi ³²p for one experiment (RhoB wild-type and knockout cells

under basal and UV or MMS conditions). To repeat this experiment twice, we need 3 X 30 mCi ^{32}P , which is way above the limit our license is permitted.

Fig. 2. Detection of R-Ras and Rheb-bound nucleotides following metabolic labeling of the cellular guanine nucleotide pool. In (A), 293T cells were transfected with plasmids encoding hemagglutinin (HA)-tagged R-Ras or the putative GEFs, GRP3 or AND-34. Following metabolic labeling of the guanine nucleotide pools with ^{32}P , HA-R-Ras was immunoprecipitated and bound nucleotides separated by TLC. Signal was detected using an AMBIS β scanner (30 min). In (B), endogenous Rheb was immunoprecipitated from two human tumor cell lines that had been metabolically labeled for 4 h with 5 mCi ^{32}P . Signal was detected by exposure to X-ray film for 6 days with intensifying screen. The percentage of GTP-bound Ras protein is shown below each lane.

(Adapted from *Methods* 2005, 37, 190–196)

Although we could not directly detect the activity of Rheb, it is well accepted that Rheb activity is reflected by mTORC1 activity in phosphorylating S6K. In fact, we have already presented that the phosphorylation of S6K is decreased after UV/MMS treatment in Figure 7A, indicating that Rheb activity is inhibited after UV/MMS treatment. In addition, Chk1 inhibitor dramatically blocked down-regulation of phosphorylation of S6K, indicating that UV/MMS treatment-induced inactivation of mTORC1 and Rheb requires Chk1 activity. Meanwhile, we added more detailed description about the relationship among TSC2, Rheb, mTORC1, and S6K in our revised manuscript.

Q3. Show effects on mTORC1 activity in UV/MMS treated cells with PIAS1 knockdown or Chk1 inhibitor.

Response: As suggested by the reviewer, we examined the effects of PIAS1 knockdown and Chk1 inhibitor on mTORC1 activity. As shown in revised Supplementary Fig.7b and 7l, both PIAS1 knockdown and Chk1 inhibitor significantly blocked UV/MMS treatment-induced inactivation of mTORC1.

Q4. Show shRNA-mediated knockdown efficacy of ATR, Chk1 and Chk2.

Response: The shRNA knockdown efficacy of ATR or Chk1 of has been provided in our revised manuscript (revised Supplementary Fig. 5d and 5g). We used Chk2 inhibitor but not shRNA for Chk2 in our experiment to discriminate which one is responsible for UV/MMS-induced autophagy (Fig. 5c), and we therefore did not check the shRNA knockdown efficacy of the Chk2.

Q5. Quantify the microscopy images of the key colocalization experiments.

Response: According to the reviewer's suggestion, the quantification results were provided for colocalization of RhoB/LAMP1, TSC2/RhoB, and TSC2/LAMP1 (revised Supplementary Fig.2, 6, and 7).

Q6. In many instances, HeLa and HEK293 cells have been used for experimentation in random order. Describe the choice of cell types and the reason for switching between them for various experiments.

Response: We used HEK293T cells only for examining proteins exogenously expressed. For all experiments related to detecting endogenous proteins, we used HeLa cells. In addition, we used U2OS cells in Fig. 2d, Fig. 6h, and Supplementary Fig. 6i to examine the subcellular localization of HA-tagged RhoB and RhoB mutants because U2OS cells have lower levels of endogenous RhoB compared with HeLa cells.

Q7. To ascertain the disease relevance, can the mechanism proposed here be demonstrated in any pathological conditions such as cancer associated with DNA damage?

Response: The reviewer is definitely right that ascertaining disease relevance would be a great addition. Our study presented that RhoB is phosphorylated by Chk1 after DNA damage, and subsequently sumoylated by PIAS1, thereby leading to a translocation of TSC2 to lysosome to initiate autophagy. Unfortunately, we do not have a specific antibody to detect the phosphorylation of RhoB yet, and it is technically impossible to examine sumoylation of RhoB in patient tumor sample. We therefore are unable to exert experiment to inspect the disease relevance to our mechanism.

Q8. Discussion section should describe why phosphorylation of RhoB is required for dissociation from plasma membrane, and why sumoylation of RhoB is required for lysosomal translocation?

Response: Following the reviewer's suggestion, we described the potential roles of phosphorylation and sumoylation of RhoB in the first paragraph of discussion section (highlighted).

REVIEWERS' COMMENTS:

Reviewer #1 (Remarks to the Author):

The authors addressed all my previous issues and improved the manuscript.

Reviewer #2 (Remarks to the Author):

The authors have satisfactorily addressed the comments with new data and explanation. Although they could not directly demonstrate any disease relevance of their study such as in cancer, they have addressed another reviewer's comments on the physiological outcome of this study. Their new data show that RhoB-mediated autophagy has a role in cell death after DNA damage. Overall, the manuscript is considerably improved over its initial version which was also very good. Therefore, it can be accepted for publication after minor correction suggested below.

Fig. S7b: The labelling of this figure seems to be incorrect. In the 4th and 7th lanes of the western blot, the control shRNA condition should have been treated with UV or MMS.

Reviewers' comments:

Reviewer #1 (Remarks to the Author):

The authors addressed all my previous issues and improved the manuscript.

Reviewer #2 (Remarks to the Author):

The authors have satisfactorily addressed the comments with new data and explanation. Although they could not directly demonstrate any disease relevance of their study such as in cancer, they have addressed another reviewer's comments on the physiological outcome of this study. Their new data show that RhoB-mediated autophagy has a role in cell death after DNA damage. Overall, the manuscript is considerably improved over its initial version which was also very good. Therefore, it can be accepted for publication after minor correction suggested below.

Fig. S7b: The labelling of this figure seems to be incorrect. In the 4th and 7th lanes of the western blot, the control shRNA condition should have been treated with UV or MMS.

Response: Thank you for the careful reading. The figure was indeed mislabeled, and we have made the changes in the finalized manuscript.